

# A New Conceptual Model for Adiabatic Fog

Felipe Toledo[1], Martial Haeffelin[2], Eivind Wærsted[3], and Jean-Charles Dupont[4]

[1]Laboratoire de Météorologie Dynamique, École Polytechnique, Institut Polytechnique de Paris, 91128 Palaiseau, France
[2]Institut Pierre Simon Laplace, École Polytechnique, CNRS, Institut Polytechnique de Paris, 91128 Palaiseau, France
[3]Laboratoire de Météorologie Dynamique, École Polytechnique, Institut Polytechnique de Paris, 91128 Palaiseau, France.
*Current affiliation:* The Norwegian Meteorological Institute, Henrik Mohns Plass 1, 0313, Oslo, Norway
[4]Institut Pierre-Simon Laplace, École Polytechnique, UVSQ, Université Paris-Saclay, 91128 Palaiseau, France

**Correspondence:** Felipe Toledo (ftoledo@lmd.polytechnique.fr)

**Abstract.** We propose a new paradigm to describe the temporal evolution of continental fog layers. This paradigm defines fog as a layer saturated from the surface to a known upper boundary, and whose liquid water path (LWP) exceeds a critical value, the critical liquid water path (CLWP). When the LWP is less than the CLWP the fog water cannot extend all the way to the surface, leading to a surface horizontal visibility greater than 1 km. On the opposite, when the LWP is larger than the CLWP,

the fog water extends all the way to the surface, inducing a horizontal visibility less than 1 km. The excess water with respect to the critical value is then defined as the reservoir liquid water path (RLWP).

    The new fog paradigm is formulated as a conceptual model that relates the liquid water path of adiabatic fog with its thickness and surface liquid water content, and allows the critical and reservoir liquid water paths to be computed. Both variables can be tracked in real time using vertical profiling measurements, enabling a real time diagnostic of fog status.

The conceptual model is tested using data from seven years of measurements performed at the SIRTA observatory, combining cloud radar, microwave radiometer, ceilometer, scatterometer and weather station measurements. In this time period we found 80 fog events with reliable measurements, with 56 of these lasting more than three hours.

    The paper presents the conceptual model and its capability to derive the LWP from the fog CTH and surface horizontal visibility with an RMS uncertainty of $10.5$ g m$^{-2}$. The impact of fog liquid water path and fog top height variations on fog life

cycle (formation to dissipation) is presented based on four case studies, and statistics derived from 56 fog events. Our results show in particular that the reservoir liquid water path is consistently positive during the mature phase of the fog and that it starts to decrease quasi monotonously about one hour before dissipation, reaching a near-zero value at the time of dissipation. The reservoir liquid water path and its time derivative could hence be used as an indicator for life cycle stage and support short range forecasting of fog dissipation.

# 1 Introduction

Fog occurs due to multiple processes that lead to saturation of the air near the surface, through cooling of air temperature, such as radiative cooling, turbulent heat exchange, diffusion, adiabatic cooling through lifting, advection, and through moistening of the air, such as evaporation from the surface, evaporation of drizzle, advection of moist air, and vertical mixing (Brown and





Roach, 1976; Gultepe et al., 2007; Dupont et al., 2012). Similarly fog dissipates as a result of warming and drying of the air
near the surface, and also through the removal of droplets by precipitation (Brown and Roach, 1976; Haeffelin et al., 2010;
Wærsted et al., 2017, 2019).

Stable fog and adiabatic fog should be distinguished because radiative, thermodynamic, dynamic and microphysical pro-
cesses are significantly contrasted in the two types of fog. In a stable fog layer, the equivalent potential temperature increases
with height, which inhibits vertical mixing. The surface is therefore weakly coupled with the fog top. Stable fog remains shal-
low and contains small amounts of liquid water, limiting the radiative cooling of the fog layer. In contrast, in an adiabatic fog
the stability is close to neutral, enabling rapid vertical mixing, so that the surface and fog top are strongly coupled (Price, 2011;
Porson et al., 2011). An adiabatic fog behaves similarly to stratocumulus clouds on top of convective boundary layers (Cermak
and Bendix, 2011). The processes of stratocumulus clouds have been studied extensively in the past with large-eddy simulation
(LES) and numerical weather prediction (NWP) models (Nakanishi, 2000; Porson et al., 2011; Bergot, 2013, 2016; Wærsted
et al., 2019).

An adiabatic fog or stratiform cloud cools at its top from emission of long wave radiation, which destabilises the cloud
and leads to convective mixing. When the cloud is coupled with the land surface, the destabilising process can be further
strengthened by heat fluxes from below due to soil heat (Price, 2011). A thermal inversion develops right above the cooling
cloud fog top and limits the coupling between the cloud and free atmosphere above. The thermal inversion defines the upper
boundary of the adiabatic fog. The lower boundary of the stratiform cloud layer varies in time and space depending the amount
of liquid water present in the cloud. For the adiabatic fog, the lower boundary is defined by the surface and is therefore fixed.
Hence a fog layer may not grow geometrically deeper when the amount of liquid water increases.

Cermak and Bendix (2011) define fog and stratiform clouds based on cloud layer top altitude and liquid water content that
follows a sub-adiabatic profile. A fog layer is thus defined as a stratiform cloud that contains sufficient liquid water to reach
down to the surface.

Wærsted et al. (2019) showed using a large eddy-simulation model and remote sensing measurements that dissipation of fog
can occur due to both reduction of liquid water content of the fog layer and increase of fog top height. Dissipation is defined
here as removal of fog droplets leading to visibility increasing above 1 km at screen-level height. The simulations reveal a
similar behavior as proposed by Cermak and Bendix (2011). For a given fog top height, if the liquid water path contained in the
fog layer becomes insufficient, the fog base lifts from the ground, which can be interpreted as fog dissipation through lifting
into a stratiform cloud.

In adiabatic clouds, the thickness can be approximated from liquid water path. Brenguier et al. (2000) state that liquid water
path is proportional to the square of cloud thickness. A precise quantification of the relationship between fog thickness and fog
liquid water path is lacking in the literature.
In this article we present a conceptual model that relates the liquid water path of adiabatic fog to its geometrical thickness
and surface liquid water content. The conceptual model enables an estimation of the minimum amount of column liquid water
that is necessary to reach a visibility less than 1000 meters at the surface, defined as the critical liquid water path, and a
calculation of the excess water that enhances fog persistence, defined as the reservoir liquid water path. The model also enables



a quantification of the impact of liquid water path and geometrical thickness variations on the reservoir, a characteristic that
could be later used to improve fog forecasting tools.

The conceptual model theory is explained in Section 2. In Section 3, we present all measurements used to construct and evaluate the conceptual model. In Section 4 we derive a parametrization for fog adiabaticity using historical data, and we compare the conceptual model predictions with fog thickness, liquid water path and surface liquid water content observations. In Section 5 we present case studies to exemplify how conceptual model variables enable us to understand fog evolution, and
statistical results of fog behavior during its formation, middle life and dissipation phases.

## 2   Fog Conceptual Model

### 2.1   Fog LWP Conceptual Model

The hypothesis of this work is that there is a general rule, dependent on fog macroscopic properties, that must be fulfilled when the fog layer persists at the surface level. To deduce this rule we develop a unidimensional model for a fog column, based on
previous models for stratus clouds.

For stratus clouds, cloud Liquid Water Content (LWC) increases with height can be modelled using Eq. (1) (Albrecht et al., 1990). In this equation, $z$ is the vertical distance above the Cloud Base Height (CBH), which increases until reaching the Cloud Top Height (CTH). $\Gamma_{ad}(T,P)$ is the negative of the change in saturation mixing ratio with height for an ideal adiabatic cloud, and $\alpha(z)$ is the local adiabaticity, defined as the ratio between the real and the ideal adiabatic liquid water content change with
height. $\Gamma_{ad}(T,P)$ is a quantity that depends on the local temperature $T$ and pressure $P$. The equation used for its calculation can be found in appendix A.

$$\frac{dLWC(z)}{dz} = \alpha(z)\,\Gamma_{ad}(T,P) \tag{1}$$

This model can also be applied for well mixed fog layers, where the adiabatic profile assumption is valid. This happens when fog is opaque, and thus radiative cooling happens almost exclusively at the fog top. This cooling introduces instability,
enhancing vertical mixing due to convective turbulence. During day time, convection is reinforced by sensible heat release from the surface. This mixing induces the formation of a saturated adiabatic temperature profile in fog layers (Roach et al., 1976; Boutle et al., 2018; Wærsted et al., 2019).

However, there is one key difference in fog layers that must be considered when integrating (1). In stratus clouds, it is assumed that the LWC at the cloud base is zero, because condensation is starting gradually from unsaturated air, and therefore
there is a smooth transition between dry and moist air.

This smooth transition does not hold in the case of fog layers. Here, CBH is not limited by the dry-moist air transition but rather by a solid boundary, the surface. The surface limits vertical fog development, and causes an excess LWC at the fog base when compared to a cloud. This larger LWC is what drives the visibility reduction at the surface. Thus, when integrating Eq. (1) it is necessary to account for a non-zero Surface Liquid Water Content ($LWC_0$). Since fog (and stratus clouds) are shallow,





the LWC increases with height, and $\Gamma_{ad}(T,P)$ can be assumed constant for the whole layer (Albrecht et al., 1990; Braun et al., 2018). This leads to the LWC formulation of Eq. (2).

$$LWC(z) = \alpha(z)\,\Gamma_{ad}(T,P)\,z + LWC_0 \tag{2}$$

The blue curve of Fig. 1 (a) illustrates how LWC behaves in well mixed fog. For most of the fog layer thickness, LWC increases with height due to upward motions of moisture from the surface and within the cloud (Oliver et al., 1978; Manton,

1983; Walker, 2003; Cermak and Bendix, 2011). Then, when approaching fog top from below, the LWC change with height decreases until becoming a net reduction of LWC near the top. This decrease is due to entrainment of dry-air at the top, which leads to a quick decline in droplet size and LWC (Brown and Roach, 1976; Roach et al., 1982; Driedonks and Duynkerke, 1989; Hoffmann and Roth, 1989; Boers and Mitchell, 1994; Cermak and Bendix, 2011).

Fog LWP is defined as the integral of LWC(z) in the fog column. Its formulation is presented in Eq. (3a), where $z$ is the

height above the surface. Since in fog the CBH is always at the surface, fog thickness is completely defined by its CTH.

$$LWP = \int_{z=0}^{z=CTH} \left( \alpha(z)\,\Gamma_{ad}(T,P)\,z + LWC_0 \right) dz \tag{3a}$$

$$LWP = \frac{1}{2}\alpha_{eq}\,\Gamma_{ad}(T,P)\,CTH^2 + LWC_0\,CTH \tag{3b}$$

To simplify the calculation of the integral in Eq. (3a), which requires the knowledge of the adiabaticity profile, we introduce the Equivalent Adiabaticity $\alpha_{eq}$ term. The Equivalent Adiabaticity is defined as a constant adiabaticity value that would provide

the same LWP as that derived from Eq. (3a). In our study, $\alpha_{eq}$ is estimated using a parametrization derived from 7 years of fog observations at the SIRTA observatory (see Sect. 4.1). It is worth mentioning that this parameter is also defined in literature as the in-cloud mixing parameter $\beta$ (e.g. Cermak and Bendix (2011)), which is equivalent to $\alpha_{eq}$ and can be easily transformed using the rule $\alpha_{eq} = (1 - \beta)$.

The Equivalent Adiabaticity enables the definition of the Fog LWP Conceptual Model, indicated in Eq. (3b). The Conceptual

Model LWP has the same value as Fog LWP, but its LWC(z) profile is different because it uses a constant adiabaticity value.

The relationship between Fog and Conceptual Model LWP is illustrated in Fig. 1 (a). Fog LWP is the light blue surface, bound by the fog LWC curve with varying adiabaticity with height. Whereas, the Conceptual Model LWP corresponds to the dashed area. Its LWC increases linearly with height because of the constant adiabaticity value. This figure shows that both Fog and the Conceptual Model have the same Surface LWC for a given LWP value. Considering that surface LWC can be linked

to visibility, this implies that for a given fog LWP value, the Conceptual Model should predict realistic visibility values at the surface.





## (a) Relationship between Fog and Conceptual Model LWC and LWP

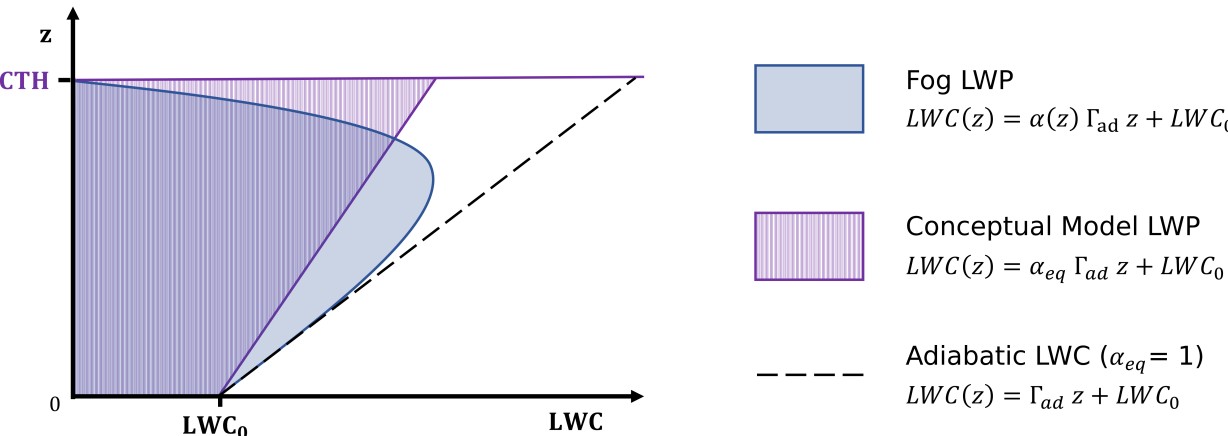

## (b) Conceptual Model Critical and Reservoir LWP

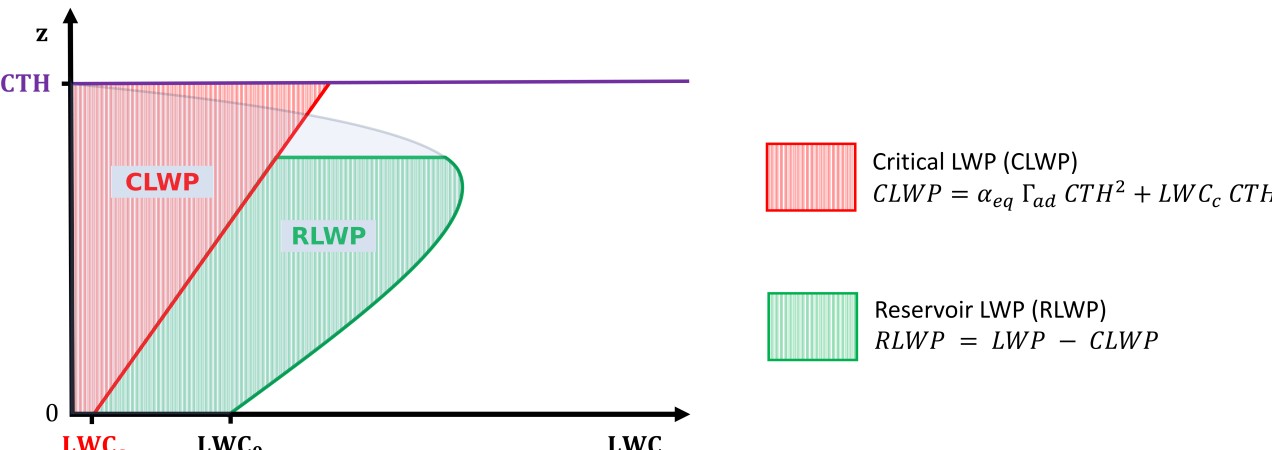

**Figure 1.** (a) Illustration of the relationship between Fog, Conceptual Model and adiabatic LWC with vs height. In all cases LWC changes with height from its surface value until reaching fog top (CTH). Fog and Conceptual Model LWP have the same value. (b) Representation of the Critical LWP (CLWP) and Reservoir LWP (RLWP) with respect to fog LWP. CLWP is predicted LWP value that fog should have when visibility equals 1000 meters at the surface (with an associated suface LWC defined as $LWC_c$). RLWP is the difference between fog and the CLWP, and represents the excess water that enables fog persistence.





## 2.2 Critical and Reservoir LWP

Wærsted (2018) found that fog dissipation by lifting of its base is explained by a deficit in LWP considering a given fog thickness. This motivated the definition of a Critical Liquid Water Path (CLWP), which is the minimum amount of LWP needed for a cloud to reach the surface, and reduce horizontal visibility below 1000 meters.

CLWP is formulated from Eq. (3b), assuming a Critical Liquid Water Content $LWC_c$ at the surface. $LWC_c$ is the LWC that would cause a 1000 meters visibility, calculated using the parametrization derived by Gultepe et al. (2006) (appendix B). This parametrization indicates that the $LWC_c$ has a value of $\approx 0.02 \; gm^{-3}$.

$$CLWP = \frac{1}{2}\alpha_{eq}\,\Gamma_{ad}(T,P)\,CTH^2 + LWC_c\,CTH \tag{4}$$

When fog is present, its LWP value must be always larger than the CLWP. This property motivates the definition of an additional parameter, the Reservoir Liquid Water Path (RLWP). RLWP is a quantitative metric on how far fog is from dissipation, and is calculated using Eq. (5).

$$RLWP = LWP - CLWP = LWP - \frac{1}{2}\alpha_{eq}\,\Gamma_{ad}(T,P)\,CTH^2 - LWC_c\,CTH \tag{5}$$

The relationship between CLWP and RLWP is illustrated in Fig. 1 (b). In this case, we have a fog with a given cloud top height CTH and a liquid water content LWP, that are associated to a liquid water content $LWC_0$ at the surface. This LWC is greater than the critical value $LWC_c$, because visibility is less than 1000 m. The CLWP of this fog, indicated by the red surface to the left, is calculated using Eq. (4). Its value indicates the minimum LWP that fog can have before reducing surface LWC below its critical value, which could cause an increase of visibility above 1000 meters. All excess liquid water above the CLWP value creates the RLWP, indicated by the green surface to the right, and corresponds to all the excess LWP that must be removed before fog can dissipate at the surface.

## 3 Dataset and Data Treatment Methodology

The dataset used to study the Conceptual Model formulation consists on seven years of fog observations made at the SIRTA atmospheric observatory, from July of 2013 to March of 2020 (Haeffelin et al., 2005). This observatory is located 156 m above sea level, approximately 20 km south of Paris (48°43'N, 2°12'E) in a location with a relatively high fog incidence (about 30 fog events per year).

The observatory data must be treated to transform raw measurements into Conceptual Model variables. Section 3.1 indicates which instruments are used in this study, Sec. 3.2 describes how fog events are detected, and how their formation and dissipation time is identified, and Sec. 3.3 explains the processing of raw observations into Conceptual Model variables.





After data treatment, an additional data quality control stage is performed to remove from the data pool the fog cases with
measurements taken under non optimal conditions. The criteria used is explained in Sec. 3.4. A summary of the complete data
processing is shown in Fig. 2.

## 3.1   Observations

The SIRTA observatory is equipped with a large array of instruments, tailored for observing fog and fog processes (Haeffelin
et al., 2010; Wærsted, 2018). A subset of these instruments is selected for studying the proposed conceptual model, based on
the required inputs. These instruments are listed in Table 1.

Data from three remote sensing instruments is used: a CL31 Ceilometer, a BASTA Cloud Radar and a HATPRO Microwave
Radiometer. The CL31 is a widely used instrument for Cloud Base Height (CBH) detection, with a vertical resolution of 15
meters (Kotthaus et al., 2016). In this study it is used to retrieve the CBH of low stratus clouds preceding fog events, and to
track CBH lifting during temporary or definitive dissipation of the fog layer.

The Cloud Radar BASTA is a 95 GHz FMCW radar used to retrieve vertical profiles of cloud reflectivity, up to 12 km
of height (Delanoë et al., 2016). It operates continuously alternating between 12.5, 25 and 100 m resolution modes every 12
seconds. The 12.5 m mode has the highest vertical resolution and therefore it is used to retrieve fog CTH. Meanwhile, the 100
m mode is the most sensitive and reaches the highest altitude of 12 km, and therefore is used to detect the presence of clouds
above the fog layer.

The multi-wavelength microwave radiometer (MWR) HATPRO measures the integrated LWP of the atmospheric column.
The manufacturer specified uncertainty of the LWP product is of $\pm$ 20 g m$^{-2}$, but for relatively small LWP ($<$ 40 g m$^{-2}$),
investigations indicate that the uncertainty is within $\pm$ 5-10 g m$^{-2}$, at least when the fog forms in clear sky so that a possible
time-independent bias can be corrected for (Marke et al., 2016; Wærsted et al., 2017). When no other cloud is present above
the fog layer, LWP measured by the MWR will correspond to fog LWP. Thus, MWR and Cloud Radar data can be combined
to perform reliable fog LWP retrievals.

These remote sensing instruments are complemented by a weather station 2 meters above the surface, and two Scatterom-
eters, at 4 and 20 meters above the surface. The weather station provides the thermodynamic data necessary to calculate the
saturated adiabatic lapse rate $\Gamma_{ad}(T,P)$, and the 4-m scatterometer provides the visibility data used to detect fog events and
to calculate fog LWC at the surface. Visibility data is also used to complement the CL31 CBH estimation for very low cloud
layers.

## 3.2   Fog event detection

Fog periods are identified using a scheme based on previous work done by Tardif and Rasmussen (2007); Wærsted et al. (2019).
This method requires the re-sampling of the surface visibility time series to 5 minute blocks. Each 5 min block is assigned a
"fog" or "clear" value, depending on the distribution of visibility in its time period. A block is assigned the "fog" value when
more than half of the visibility measurements are less than 1000 m, and is assigned "clear" otherwise.





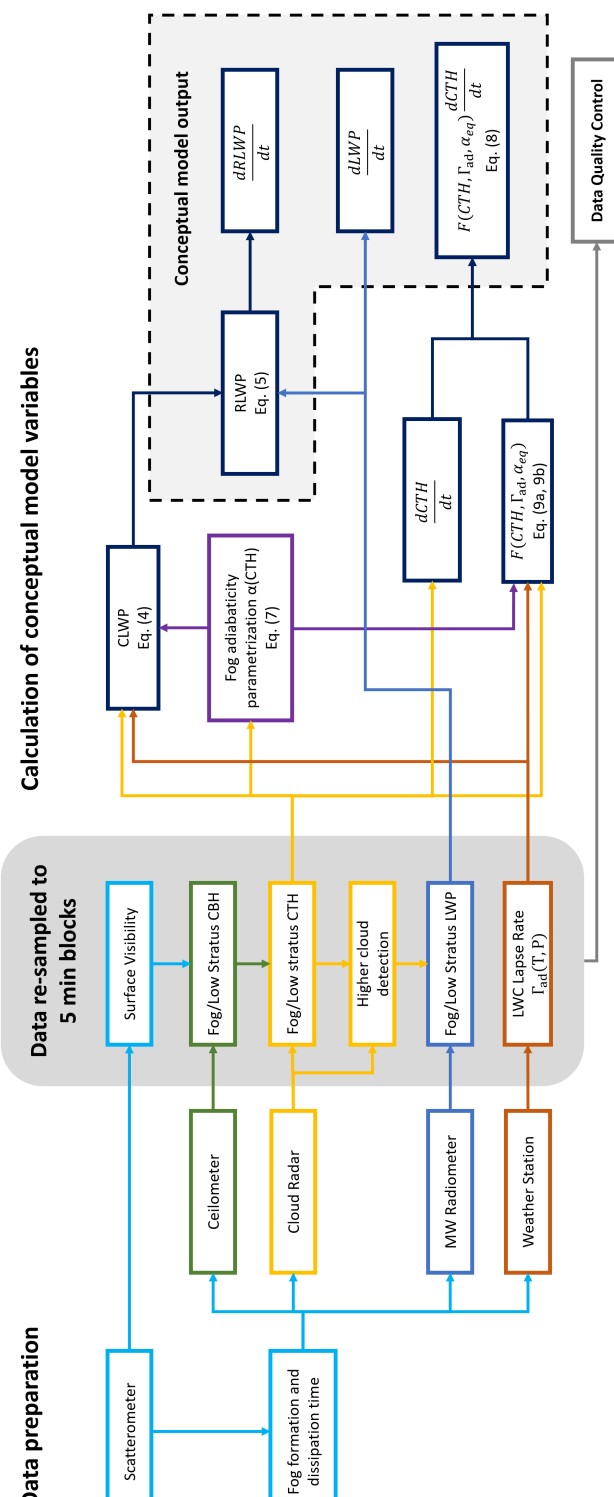

**Figure 2.** Summary of the data treatment and calculation methodology. The procedure can be separated in three main stages: First, data preparation consists in identifying fog periods from historical visibility measurements, and in gathering raw instrumental information for these periods. Second, data is re-sampled and homogenized into 5 minute time blocks. First order products such as fog CTH, LWP, among others are calculated. Third, the data treated in the second stage is used to calculate conceptual model variables. An additional data quality control stage is included, to check if fog variables of each identified period were retrieved under reliable instrument operation.



**Table 1.** List of instruments and measurements used in this study.

| Instrument | Measured Quantity | Vertical Range and Resolution | Time Res. |
|---|---|---|---|
| 905 nm Ceilometer *Vaisala CL31* | Attenuated backscatter $(m^{-1} \, sr^{-1})$ | RA 0-7600 m, RE 15 m | 60 s |
| 14-Ch. Microwave Radiometer *RPG HATPRO* | Liquid Water Path $(g \, m^{-2})$ | Integrated column | 60 s |
| 95 GHz FMCW Cloud Radar *BASTA* | Radar Equivalent Reflectivity (dBZ) | RA 85-6000 m, RE 12.5 m RA 100-12000 m, RE 100 m | 12 s 12 s |
| 550 nm Scatterometer *Degreane DF320/DF20+* | Visibility (m) | At 4 m At 20 m | 60 s 60 s |
| Thermometer *Guilcor PT100* | Air Temperature (K) | At 2 m | 60 s |
| Barometer *Druck RPT410F* | Surface Pressure (Pa) | At 2 m | 60 s |

After asigning values to each block of the complete visibility time series, we analyze groups of five consecutive blocks in a sliding manner. These five contiguous blocks are defined as a construct, and its value is positive when the central and at least two other are fog blocks, and negative otherwise.

A fog event forms when a positive construct is encountered, with a formation time defined as the central time of the first fog block in the construct. Conversely, a fog event dissipates when the last positive construct is followed by either a negative construct or three consecutive clear blocks. Fog dissipation time is set as the central time of the block immediately after the last fog block in the last positive construct. Fog events separated by less than 1 hr are merged, and all fog events lasting less than 1 hr are discarded. This algorithm provides the formation and dissipation time of 217 fog events between July 2013 and March 2020.

## 3.3 Data processing

After identifying the fog events, it is necessary to process raw measurements from the instruments into information that can be used by the conceptual model. To enable the study of the conceptual model variables during fog itself and the time period surrounding it, observational data is automatically processed from 3 hours before fog formation to 3 hours after fog dissipation.

CBH is retrieved using a threshold value of $2 \cdot 10^{-4} \, m^{-1} sr^{-1}$ on the CL31 attenuated backscatter measurements, following the method of Haeffelin et al. (2016). When the liquid layer is closer than 15 m to the ground, the CL31 cannot identify the CBH anymore and therefore the Scatterometer measurements are checked, setting the CBH as 0 m when visibility drops below 1000 m. Both CBH and visibility measurements are averaged to five minute time blocks, matching the blocks used by the fog detection algorithm.





The Cloud Radar is used to retrieve fog CTH and to detect the presence of higher clouds above the fog layer, based on its
vertical reflectivity profile (Wærsted et al., 2019). To retrieve CTH, reflectivity signals in each radar gate are analyzed, starting
from the gate closest to the CBH and checking one gate at a time, going upwards. CTH is estimated as the height of the gate
under the first gate where no cloud signal is detected. A gate is considered to have a valid cloud signal if more than half of the
reflectivity samples in a five minute time block are not removed by the automatic noise filtering algorithm of the radar (Delanoë
et al., 2016). As with CBH, time blocks used in CTH retrievals match those defined for fog detection.

A limitation of this method is that the minimum detectable CTH is of 85 meters. Under this height, radar interference
becomes very significant, making the differentiation between a valid cloud signal and noise very difficult. Therefore, we
decide to not use data associated with CTH retrievals below 85 meters.

Additionally, radar data treatment creates a flag indicating the possible presence of liquid clouds above the fog layer if
another valid signal is observed above fog CTH within the first kilometer for the 12.5 m resolution mode, or within the first
6000 m for the 100 m resolution mode.

The HATPRO Microwave Radiometer is used to perform LWP retrievals of fog. LWP measured by HATPRO is averaged to
the 5 min time blocks and then is filtered using the radar higher cloud flag. This is done to avoid fog LWP overestimation when
liquid clouds are present above the fog layer.

Time series of surface temperature and pressure are all averaged to match the 5 minute time blocks. The saturated adiabatic
lapse rate $\Gamma_{ad}(T,P)$ is calculated for each of these time blocks using these measurements and the equations in appendix A.

### 3.4 Data quality control

After data treatment is complete for all automatically detected fog events, a manual check is done to remove cases where data
is unreliable. This happens when instruments operate under non optimal conditions, or when the upper liquid cloud flagging
algorithm did not work correctly.

This control consist on accepting or removing complete fog cases and their associated dataset. A fog case is removed from
the data pool if measurements taken when the fog takes place comply with at least one of the following criteria:

1. Data is taken during or after strong precipitation: Strong precipitation wet the Microwave Radiometer radome, leading
   to unreliable LWP retrievals for an unpredictable period of time that can last up to hours, even when following all
   maintenance instructions (Görsdorf et al., 2020). Additionally, strong rain leads to difficulties in identifying the fog CTH
   because the strong reflectivity from rain hides the weaker returns from suspended fog droplets.

2. There are no valid data blocks: No CTH or LWP retrievals could be made for the given fog event. This can happen when
   fog is thinner than 85 meters, or when liquid clouds are present above fog for the complete event duration.

3. Fog and Cloud borders are not well identified: In some cases the automatic cloud border detection algorithm fails,
   leading to unfiltered LWP retrievals with liquid clouds above, or to a bad estimation of fog CTH when upper clouds
   are too close to the fog layer. The latter can be seen in the radar data as multilayer fog formed by the union of two





previously independent cloud layers. This situation departs from the single well mixed layer assumption, and therefore the conceptual model is not applicable.

The quicklooks for the accepted and rejected fog cases are available in the article supplementary material. After this stage we end with 80 valid fog cases and 137 rejected cases, where 50 were removed because of criterion 1, 69 because of criterion 230 2 and 18 because of criterion 3. These 80 valid fog cases pass to the next stage of data analysis and results.

## 4 Data Analysis and Results

### 4.1 Fog Adiabaticity

A key parameter in the calculation of the CLWP is the Equivalent Fog adiabaticity $\alpha_{eq}$ (Eq. (4)). This parameter has been previously studied in literature for boundary layer stratocumulus and stratus clouds, where typically observed values of $\alpha_{eq}$ 235 range between 0.6 and 0.9 (Slingo et al., 1982; Boers et al., 1990; Boers and Mitchell, 1994; Braun et al., 2018).

It is interesting to study whether these values also apply to fog, since fog is a special case of cloud whose vertical development is limited by a solid lower boundary at the surface. Therefore, we use our database to calculate $\alpha_{eq}$ by closure, with Eq. (6). This equation is an inversion of the conceptual model formulation of Eq. (3b), and enables an estimation of the adiabaticity correcting the impact of excessive LWC accumulation caused by the solid lower boundary. We only perform $\alpha_{eq}$ retrievals 240 when visibility is below 2000 m, in order to remain close to fog conditions, where the conceptual model is valid.

$$\alpha_{eq}^{closure} = \frac{2(LWP - LWC_0 \, CTH)}{\Gamma_{ad}(T,P) \, CTH^2} \tag{6}$$

Figure 3 (a) shows the relationship between $\alpha_{eq}^{closure}$, CTH and LWP. The results indicate that fog layers with small LWP are characterized by large ranges of $\alpha_{eq}^{closure}$, and sometimes it can even reach negative adiabaticity values.

This can be explained by considering that fog with LWP less than 30 g m$^{-2}$ is not optically thick. Under this condition, the 245 liquid water condensation happens everywhere in the liquid layer, but it is mostly driven by surface cooling (Wærsted et al., 2017). This process is associated with stable atmospheric conditions, where vertical mixing is almost neglibile (Zhou and Ferrier, 2008). Under this regime, the LWC will be distributed according to the cooling and condensation rate at each height, and therefore it is likely to have situations where surface LWC is greater than LWC values above, especially during radiation fog formation. This situation would lead to the observed negative $\alpha_{eq}$ values.

When fog LWP begins to surpass 30 g m$^{-2}$, fog gradually becomes opaque to infrared radiation (Wærsted et al., 2017). In this scenario, LWC generation is mostly driven by radiative cooling at the fog top. This radiative cooling induces a temperature gradient between the fog top and the surface, leading to convective motions. An increase in the intensity of convection will be correlated with an increase in fog CTH, because the additional energy would enhance boundary layer development. Then, as fog becomes deeper, it is expected that the relatively stronger convective motions associated would drive the vertical liquid 255 water mixing closer to what is observed in boundary layer clouds. Our results are consistent with this description. $\alpha_{eq}^{closure}$

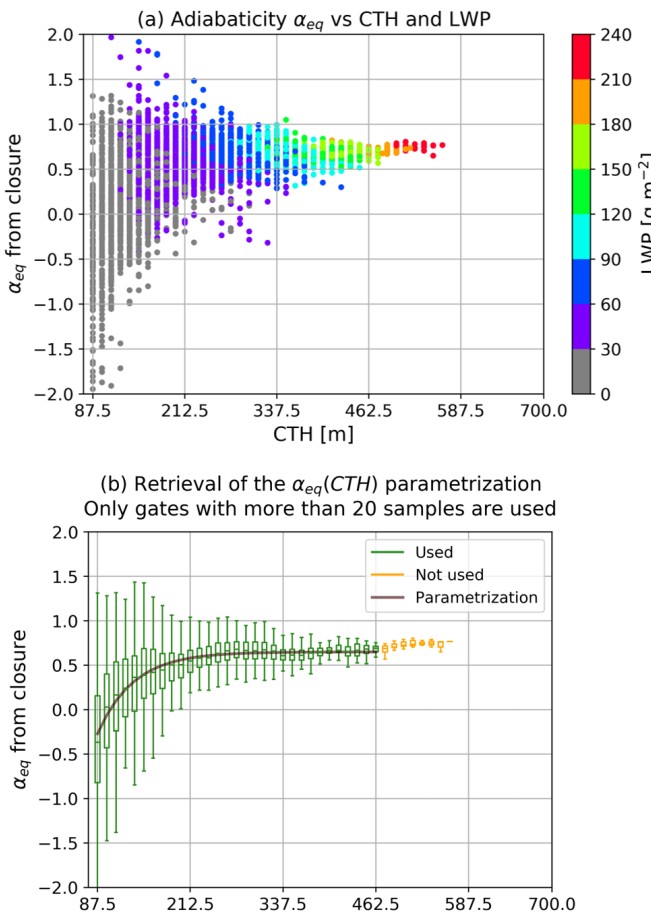

**Figure 3.** (a) Closure adiabaticity results versus fog CTH and LWP, calculated using Eq. (6) and samples with visibility inferior to 2000 m. (b) Boxplot with the distribution of closure adiabaticity for each radar bin, with the derived parametrization superimposed (Eq. (7)).

increases in average with fog CTH until reaching a plateau at $\alpha \approx 0.7$, a value consistent with typical observations of boundary layer stratocumulus (Slingo et al., 1982; Boers et al., 1990; Boers and Mitchell, 1994; Cermak and Bendix, 2011; Braun et al., 2018).

The data presented in Fig. 3 (b) suggests that $\alpha_{eq}$ can be parametrized as a function of CTH. The parametrization curve is calculated by minimizing the error of the model presented in Eq. (7) with respect to the median $\alpha_{eq}$ value at each radar range bin. To reduce uncertainty due to lack of data, we only use range bins with more than 20 valid samples.

$$\alpha_{eq}(CTH) = \alpha_0 \left( 1 - e^{-\frac{CTH - H_0}{L}} \right) \tag{7}$$




The retrieved value for each coefficient are $\alpha_0 = 0.66$, $H_0 = 107.3$ m and $L = 50.2$ m. These parameters come from fog statistical behavior, and can be interpreted as follows: $\alpha_0$ is the equivalent adiabaticity value that fog reaches when it has

completely transitioned into an adiabatic regime. $H_0$ is the usual height at which LWC starts to increase with height. $L$ indicates, based on adiabaticity, that the transition from stable to adiabatic fog is possible when CTH reaches 150 meters, and very likely when CTH is above 250 meters ($H_0 + L$ and $H_0 + 3L$ respectively).

In principle, the adiabaticity parametrization is valid for CTH values below 462.5 m, where the parametrization is derived. Beyond this height there is not enough data to guarantee its reliability; however it is likely that adiabaticity should remain close

to the convergence value of 0.66 based on our observations and on what has been previously published in literature (Slingo et al., 1982; Boers et al., 1990; Boers and Mitchell, 1994; Cermak and Bendix, 2011; Braun et al., 2018).

## 4.2 Conceptual model validation

In this section we study fog statistical data to study how it behaves with respect to the conceptual model. Figure 4 (a) shows all CTH, LWP and surface LWC measurements taken when fog is present (visibility less than 1000 m). Data is separated in

different temperature ranges. Modeled LWP and CLWP curves are shown. LWP and CLWP theoretical curves are calculated using Eqs. (3b) and (4) respectively, with the $\alpha_{eq}(CTH)$ parametrization derived in Sec. 4.1. Each hexagon color is given by the mean $LWC_0$, calculated using all the data in their respective CTH+LWP space. Hexagons with less than 5 samples within their surface are removed, since they are likely to be associated with non replicable, noisy data.

This figure shows a good agreement between the theoretical curves and observed results. Most LWP samples are higher than

the critical value, as the model predicts when visibility is less than 1000 meters. Additionally, it can be seen that for a fixed CTH, LWP increases with $LWC_0$. This behavior seems to be well captured in the current Conceptual Model formulation, as the difference between the three lines shows (each theoretical LWP line has a different $LWC_0$ value, indicated in the legend).

Figure 4 (b) shows data samples taken when visibility is between 1000 and 2000 meters, as an scatterplot. As in Sec. 4.1, the 2000 m superior limit to visibility is selected, to remain close to fog conditions where the conceptual model is valid. LWP of

these data samples should be less than the CLWP line for these visibility values, however we observe that sometimes they can also be larger. This can be explained by two main reasons: CLWP is calculated for a single temperature while data temperature varies within a range, and because of instrumental uncertainties. HATPRO LWP uncertainty is around $10 \text{ g m}^{-2}$, while radar CTH retrieval has a resolution of 12.5 m. This uncertainty is present in this retrieved data, and is also likely to be propagated inside the $\alpha_{eq}(CTH)$ parametrization, introducing some variability in the results. However this is not deemed critical, since

variability around the CLWP line is smaller than $10 \text{ g m}^{-2}$, and because the fog life cycle studies of Sec. 5) verify that LWP is lower than the critical value before fog formation and after fog dissipation.

Finally, we perform an evaluation on how well the Conceptual Model predicts fog LWP, based on CTH, Temperature, Pressure and surface LWC inputs. These variables are used to calculate the Conceptual Model LWP with Eq. (3b), with the $\alpha_{eq}(CTH)$ parametrization of Sec. 4.1, and compared against HATPRO LWP retrievals. Results are shown in Fig. 5. Here we

can see that most samples are close to the 1-1 line for LWP values less than approximately $190 \text{ g m}^{-2}$. Beyond this LWP value some deviation appears, however there is not enough data available to verify if this is a systematic error of the model or on

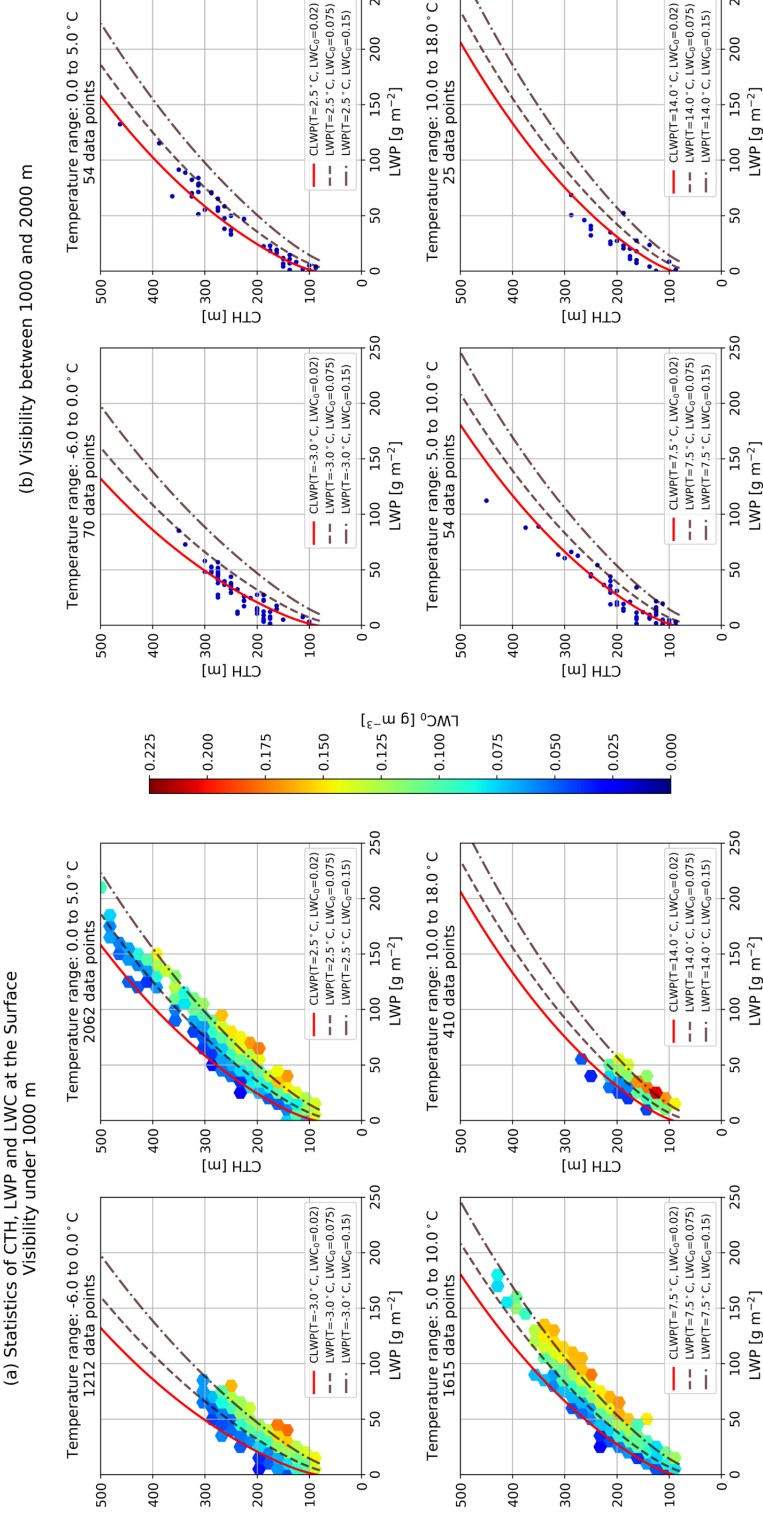

**Figure 4.** Observations of CTH, LWP and LWC at the surface for different temperature and visibility ranges. Data associated to visibility values below 1000 m is to the left (title (a)), while data measured with visibility values between 1000 and 2000 m is to the right (title (b)). Conceptual model theoretical LWP and CLWP lines for different conditions, indicated in the legend, are superimposed. The adiabaticity values used in the conceptual model calculation come from the adiabaticity parametrization of Sec. 4.1.





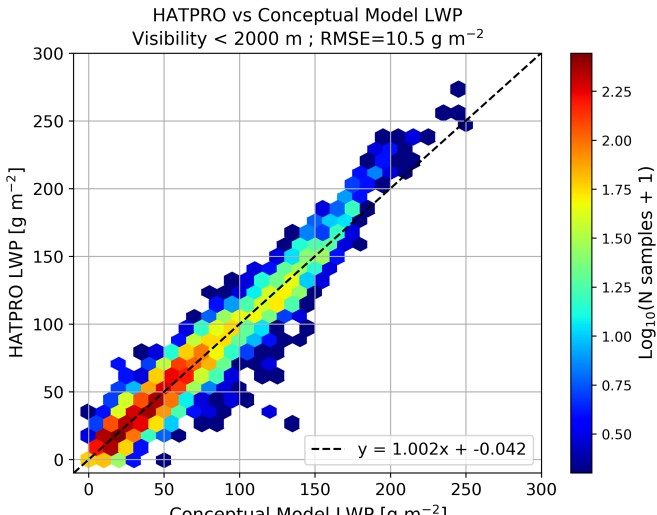

**Figure 5.** 2D histogram comparing HATPRO and Conceptual Model LWP values, for data retrieved when visibility is less than 2000 m. Conceptual model LWP is calculated using fog CTH, fog LWC at the surface derived from visibility, surface temperature, surface pressure and the adiabaticity parametrization of Eq. (7). Under these conditions, the conceptual model predicts LWP with an RMSE of 10.5 g m$^{-2}$ and an almost perfect linear relationship.

how data was taken. Despite this deviation, the good agreement between modeled and observed LWP can be seen in the linear fit, with a slope equal to 1, and in the RMSE of just 10.5 g m$^{-2}$, which is very close to the LWP retrieval uncertainty.

### 4.3 Drivers of RLWP temporal variations

Equation (5), indicates that changes in both LWP and CTH can contribute to RLWP depletion, and therefore to fog dissipation. To quantify the relative impact of LWP and CTH changes in RLWP, we calculate the time derivative of Eq. (5). By assuming constant temperature and pressure, and using the $\alpha(CTH)$ parametrization of Sec. 4.1, we obtain Eq. (8).

This equation shows that RLWP changes are proportional to LWP variations, and to CTH variations weighted by the function $F(CTH, \Gamma_{ad}, \alpha_{eq})$. This function, written explicitly in Eqs. (9a) and (9b), converts CTH variations into g m$^{-2}$ units, and thus
enables a comparison between both effects.

$$\frac{dRLWP}{dt} = \frac{dLWP}{dt} - F(CTH, \Gamma_{ad}, \alpha_{eq}) \frac{dCTH}{dt} \tag{8}$$

$$F(CTH, \Gamma_{ad}, \alpha_{eq}) = \frac{1}{2} \frac{\partial \alpha_{eq}(CTH)}{\partial CTH} \Gamma_{ad}(T, P) \, CTH^2 + \alpha_{eq}(CTH) \, \Gamma_{ad}(T, P) \, CTH + LWC_c \tag{9a}$$

$$\frac{\partial \alpha_{eq}(CTH)}{\partial CTH} = \frac{\alpha_0}{L} e^{-\frac{CTH - H_0}{L}} \tag{9b}$$





Equation (8) implies that RLWP depletion, and thus fog dissipation, can occur by LWP reduction and/or by CTH growth. It also indicates that it is possible to have compensating effects enhancing fog persistence, for example fog that is reducing its LWP could persist if its CTH is also decreasing (which can happen under strong subsidence). Another implication is that it is possible to have fog dissipation even if LWP is increasing quickly, through a fast increase in CTH. The case studies of Sec. 5.1 show how useful this separation between LWP and CTH effects can be, by analyzing some examples of the previously mentioned scenarios. Section 5.2.3 shows statistical results of fog RLWP, LWP and CTH time derivatives just before dissipation.

## 5   Fog life cycle

### 5.1   Case studies

We present 4 case studies to illustrate the behavior and role of changes in LWP and CTH on presence of fog at the surface during the fog life cycle (Figs. 6, 7, 8 and 9). For each case we provide a 6-panel figure that illustrates the time series of fog/stratus layer boundaries, reflectivity profile, 4-m and 20-m horizontal visibilities, the fog/stratus layer measured LWP and computed RLWP, temperature and closure adiabaticity; and the change rate of RLWP, with the individual contributions from LWP and CTH variations.

In all four cases, we observe that fog is present at the ground (4-m height visibility < 1 km) when the RLWP is greater than 0 g m$^{-2}$. RLWP changes at a rate of +/-10 g m$^{-2}$ Hr$^{-1}$, with values reaching + or – 30 g m$^{-2}$ Hr$^{-1}$ at times. The LWP estimation of all case studies is done directly with the HATPRO, since as radar images show, there are no cloud signals below 6 km of height (panel (b) of each case study figure).

Case study 1 (Fig. 6): Radiative fog occurring during fall season (31 October 2015) that forms six hours before sunrise and dissipates about three hours after sunrise at 10:25 UTC. The fog layer is about 200 m thick during the entire fog life cycle with a water content of 30-60 g m$^{-2}$. This LWP range and the adiabaticity values close to 0.6 indicates that fog is optically thick and can be considered as a well-mixed layer for most of its duration. The RLWP is not large, mostly near + 10 g m$^{-2}$, with a maximum value of 30 g m$^{-2}$ observed 2-3 hours before sunrise. CTH changes are relatively slow during the entire fog life cycle, with values less than 50 m Hr$^{-1}$. From 03 to 05 UTC, the CTH increases which acts as RLWP depletion of nearly -20 g m$^{-2}$ Hr$^{-1}$, while at the same time the LWP increases with a rate reaching +50 g m$^{-2}$ Hr$^{-1}$ resulting in a net increase of RLWP. After 05 UTC, the trends in CTH and LWP reverse. The CTH subsides slowly (about -20 m Hr$^{-1}$) contributing positively on the RLWP at a rate of nearly +5-10 g m$^{-2}$ Hr$^{-1}$, while the LWP initiates a progressive and nearly monotonous decrease of -10 g m$^{-2}$ Hr$^{-1}$ that brings the RLWP to 0 g m$^{-2}$ at 09 UTC. The progressive drying of the fog layer is also identifiable in the closure adiabaticity value, which starts to decrease just after sunrise. After 09 UTC, the near-surface visibility initiates a rapid increase, exceeding 1 km at 10:25 UTC, time at which the entire fog layer is dissipated. The complete layer dissipation and the increasing temperature makes it highly unlikely that fog will re-form in the coming hours. Note on Fig. 6 (f) that LWP and CTH contributions to RLWP are nearly always of opposite signs, but not equal in magnitude.





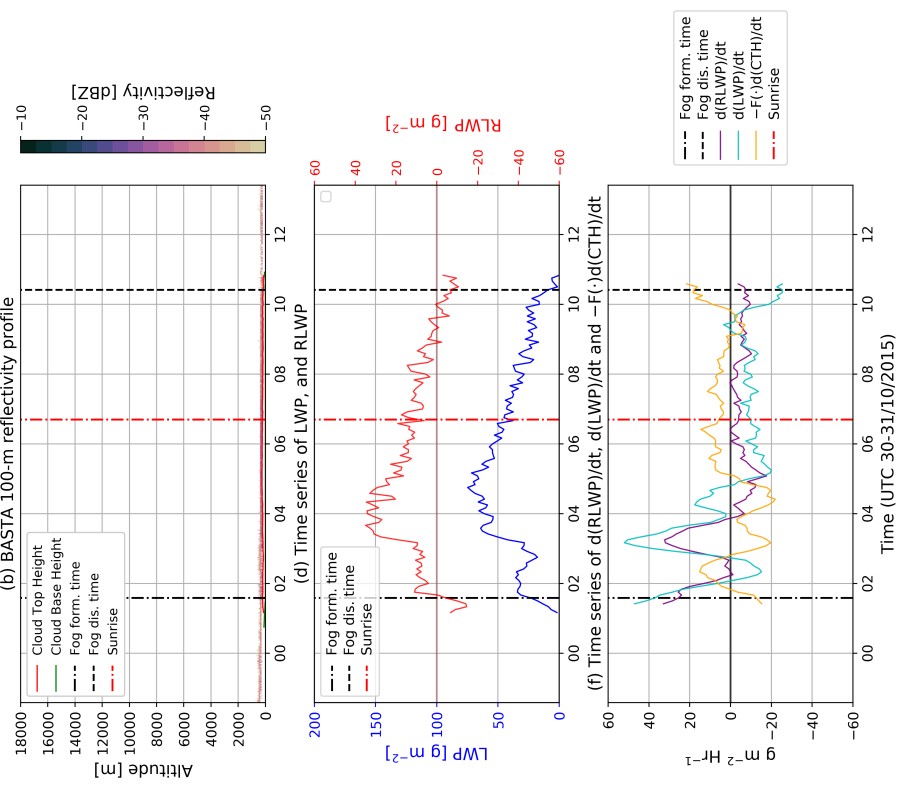

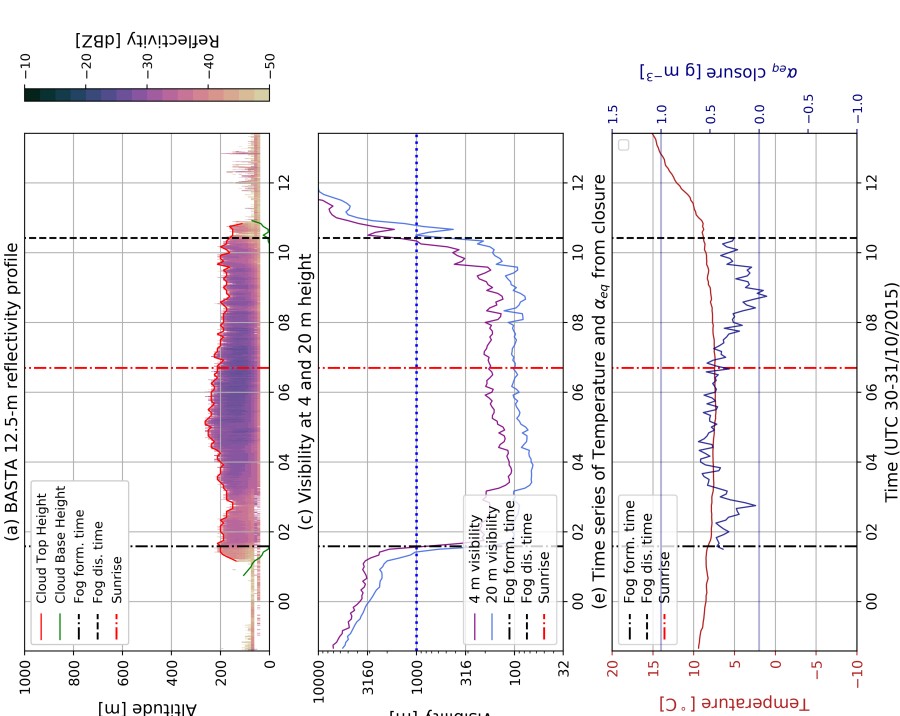

**Figure 6.** Case study 1. (a) Cloud Base Height (CBH), Cloud Top Height (CTH), and the cloud radar 12.5 m resolution reflectivity profile for the first 1000 m of height. (b) Cloud radar 100 m resolution reflectivity profile up to 10 km of height. (c) 4-m and 20-m horizontal visibilities. (d) Fog/Stratus layer measured LWP and computed RLWP. (e) Temperature and closure adiabaticity (calculated only when visibility is less than 2000 m). (f) Change rate of RLWP, with the individual contributions from LWP and CTH variations. In each panel, the time of fog formation and fog dissipation are clearly marked as well as the time of sunrise.





Case study 2 (Fig. 7): Another radiative fog that occurs in the fall season, just a few days apart from case study 1 (26 October 2015). It forms just three hours before sunrise and dissipates about 3.5 hours after sunrise at 10:55 UTC. The fog layer is about 200 m thick during the mature phase of the fog life cycle and nearly doubles between sunrise and time of dissipation, while the water content remains above 50 g m$^{-2}$. After fog formation, RLWP reaches 30 g m$^{-2}$ in about one hour and remains at

this level for about 2 hours. Fog adiabaticity indicates that after the first hour from formation fog remains in a well mixed state. Around sunrise, RLWP initiates a nearly monotonous decreasing trend of -10 g m$^{-2}$ Hr$^{-1}$ that will last until fog dissipation. The negative RLWP rate is driven by the rise of CTH that contribute negatively on RLWP with a rate that exceeds -20 g m$^{-2}$ Hr$^{-1}$ only partially compensated by +20 g m$^{-2}$ Hr$^{-1}$ LWP increase rates. Oscillations in LWP and CTH contributions to RLWP are clearly visible in Fig. 7 (f). When there is strong cooling at the fog layer top, LWP increases and vertical circulation

is intensified. This increases mixing with the layer above fog, resulting in a CTH increase. On the contrary, processes associated with CTH subsidence tend to decrease LWP rates (Wærsted, 2018). In this case study, the depletion of RLWP is clearly driven by the CTH increase and the fog LWP still exceeds 75 g m$^{-2}$ at the time of dissipation.

Case study 3 (Fig. 8): This third case occurred also in the fall season a few days apart from the two previous cases. It is characterized by a very late dissipation time 14:40 UTC, which is eight hours after sunrise. We will focus here on differences

with the two previous cases that can explain the persisting character of this fog layer. During the fog life cycle of case 3, the CTH behaves similarly to that of case 1, reaching about 200 m agl and revealing both positive and negative evolutions with a rate of about +/- 25 m Hr$^{-1}$. During the fog life cycle of case 3, the LWP behaves similarly to that of case 2, ranging between 50 and 100 g m$^{-2}$, which leads to a fog layer with a significant RLWP reaching +50 g m$^{-2}$ during the first five hours of its life cycle. Adiabaticity remains always close to 0.6, indicating that the layer is well mixed for the complete fog duration. At 12:00

UTC, nearly six hours after sunrise, the RLWP is still greater than +30 g m$^{-2}$, a clear sign that the fog is not about to dissipate, as confirmed by the 200-m near surface visibility. Between sunrise and noon, the rate of change of RLWP switches from positive to negative values (mostly within +/-10 g m$^{-2}$ Hr$^{-1}$). Contributions to RLWP changes from LWP and CTH changes are of opposite signs, with LWP being the dominant contributor, and their values cross the zero line nearly simultaneously eight times during the fog life cycle (Fig. 8 (f)). This is clear evidence of the feedback mechanisms that occur in fog between

CTH changes and LWP, that tend to dampen the evolution of RLWP and lead to fog layers that persist for many hours even during daytime. The fog dissipation at 14:40 results primarily from a significant decrease of LWP after 12:00 UTC with a comparatively stable CTH. At the time of dissipation, when RLWP reaches 0 g m$^{-2}$, there is still about 25 g m$^{-2}$ of LWP in the 200 m thick fog layer, which becomes insufficient to maintain the fog at the ground.

Case study 4 (Fig. 9): Here we have a typical case of a very low stratus cloud layer with CTH near 250 m agl and an LWP that

ranges 25-50 g m$^{-2}$. This combination leads to a negative RLWP that is insufficient for the stratus to deepen all the way to the surface. As expected for low stratus clouds, the value of closure adiabaticity is close to 0.6 for all valid samples (when visibility is less than 2000 m, to have valid conceptual model conditions with positive LWC at the surface). The stratus is present from 18:00 UTC onwards during twelve hours with a near-surface visibility of about 2-3 km. From 18 until 23 UTC, RLWP is clearly negative changing frequently from negative to positive rates of change (about +/- 5 g m$^{-2}$ Hr$^{-1}$) as the contributions

of LWP and CTH changes oscillate from positive to negative values (as also seen in Case 3). At 01 UTC, the stratus reaches



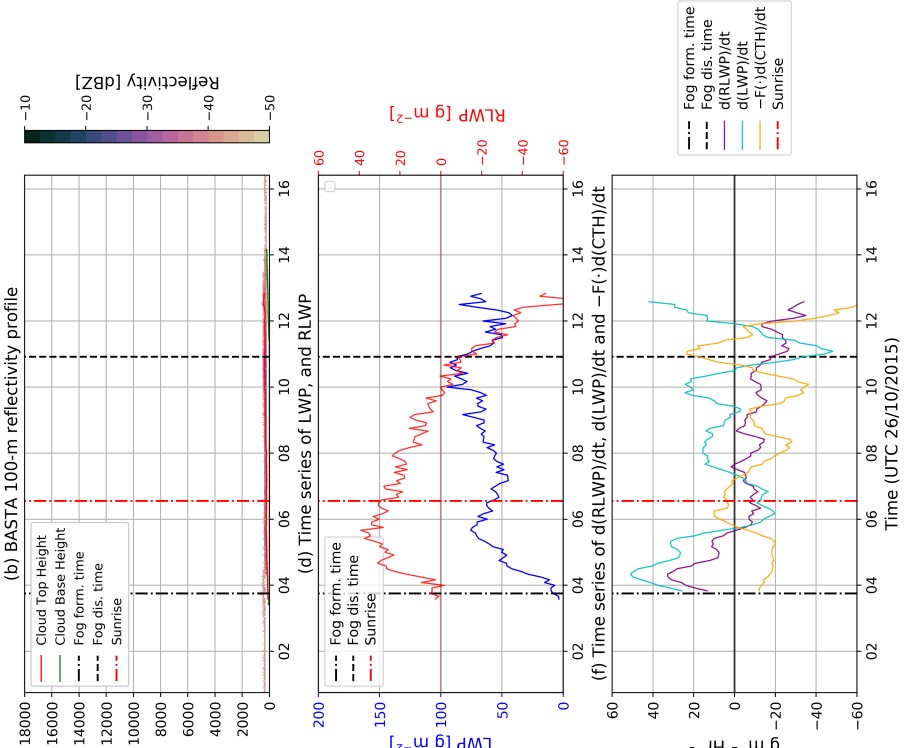

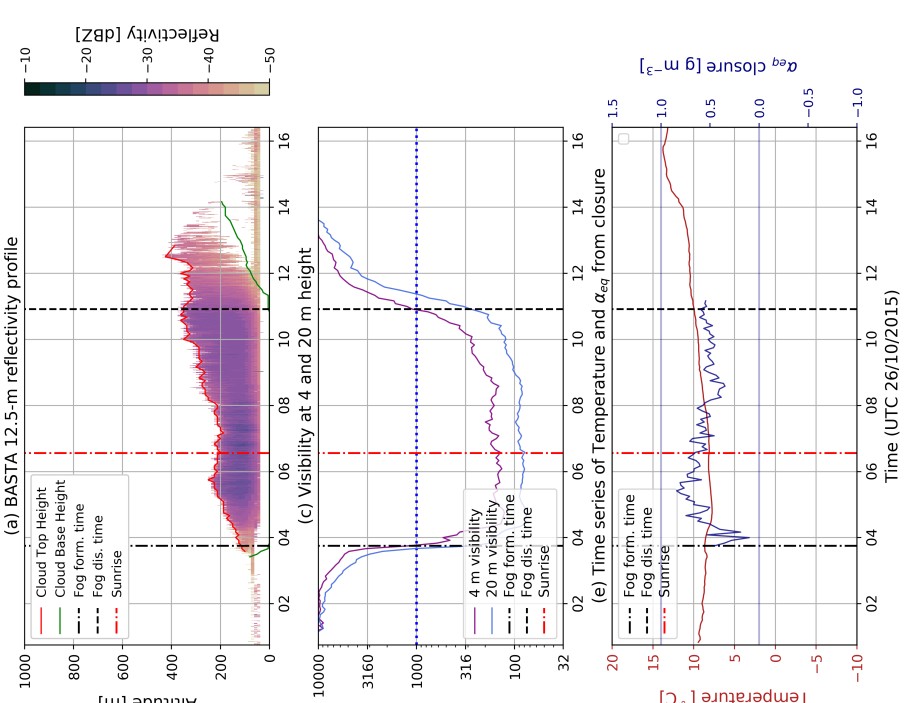

**Figure 7.** Case study 2. (a) Cloud Base Height (CBH), Cloud Top Height (CTH), and the cloud radar 12.5 m resolution reflectivity profile up to 10 km of height. (b) Cloud radar 100 m resolution reflectivity profile for the first 1000 m of height. (c) 4-m and 20-m horizontal visibilities. (d) Fog/Stratus layer measured LWP and computed RLWP. (e) Temperature and closure adiabaticity (calculated only when visibility is less than 2000 m). (f) Change rate of RLWP, with the individual contributions from LWP and CTH variations. In each panel, the time of fog formation and fog dissipation are clearly marked as well as the time of sunrise.



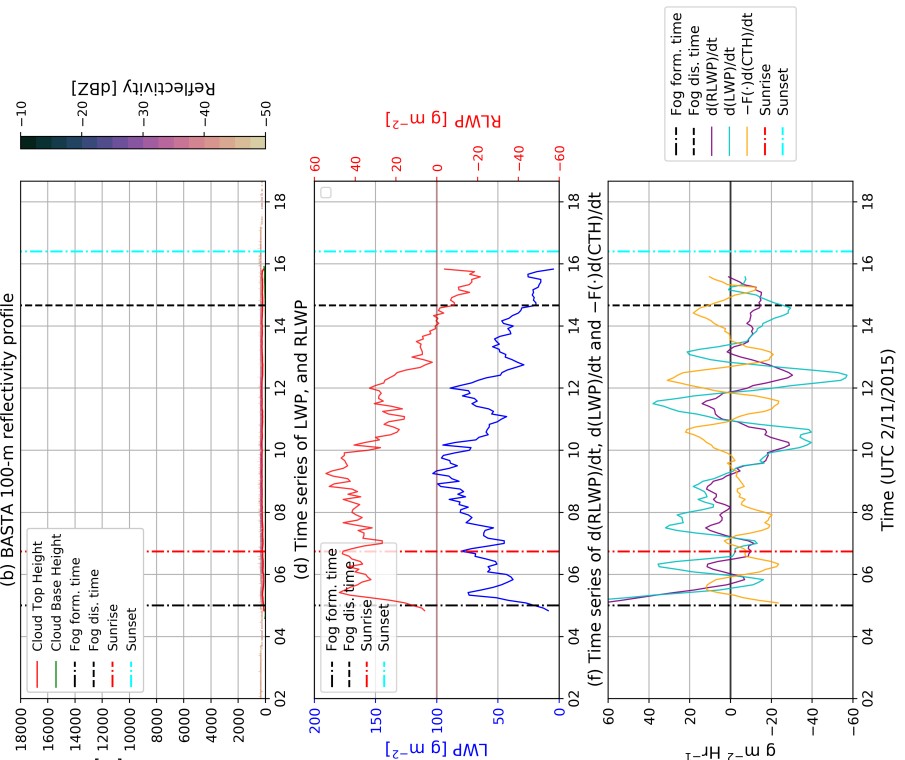

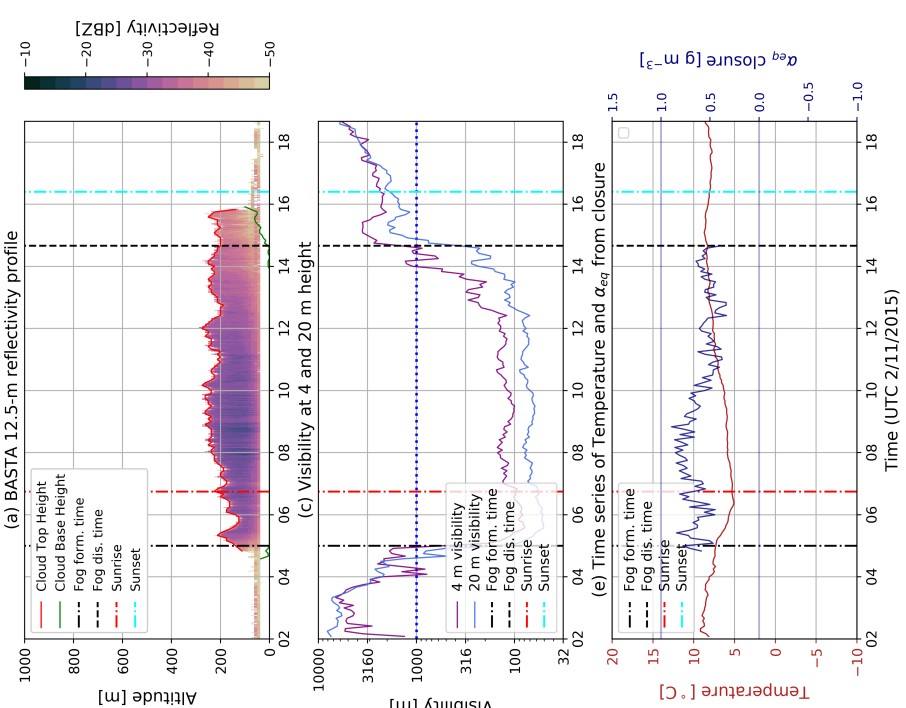

**Figure 8.** Case study 3. (a) Cloud Base Height (CBH), Cloud Top Height (CTH), and the cloud radar 12.5 m resolution reflectivity profile up to 10 km of height. (b) Cloud radar 100 m resolution reflectivity profile for the first 1000 m of height. (c) 4-m and 20-m horizontal visibilities. (d) Fog/Stratus layer measured LWP and computed RLWP. (e) Temperature and closure adiabaticity (calculated only when visibility is less than 2000 m). (f) Change rate of RLWP, with the individual contributions from LWP and CTH variations. In each panel, the time of fog formation and fog dissipation are clearly marked as well as the time of sunrise.





a new equilibrium with an LWP hovering around 50 g m$^{-2}$, which brings the RLWP very close to 0 g m$^{-2}$. The fog CBH is then below 20 agl, as evidenced by the visibility values measured at 20 m agl (Fig. 9 (c)). Between 04:30 and 06:30 UTC, the RLWP becomes again negative and the stratus base lifts. A strong increase in LWP (+40 g m$^{-2}$ Hr$^{-1}$) starting after 06:00 UTC leads to a positive RLWP after 06:30 UTC and the stratus layers deepens all the way to the surface. The trend in LWP reverses

around 08 UTC (-20 g m$^{-2}$ Hr$^{-1}$) while the CTH remains mostly constant hence reducing the RLWP towards 0 g m$^{-2}$ before 10 UTC. This case study shows that the RLWP is also a good indicator of the possibility for a very low stratus layer to deepen into fog and then reversely for the fog to lift into a low stratus.

## 5.2    Fog life cycle statistics

Taking advantage of our large database, we study the behavior of fog RLWP and its time derivative dRLWP/dt statistically, for
three different periods: fog formation, mature stage and dissipation. The objective is to identify patterns that these fog variables follow at each stage. This could lead to the development of new indicators to enhance the capabilities of fog forecasting models.

Fog formation statistics are taken between 90 minutes before and 90 minutes after the time block were fog formation is identified from visibility measurements (Sec. 3.2). Likewise, for the dissipation period the analyzed data is taken from 90 minutes before to 90 minutes after the dissipation time block. All remaining blocks between 90 minutes after fog formation,
and 90 minutes before fog dissipation, are considered to be fog middle life data. Because of how the fog stages are defined, the cases included in this statistical analysis must have a duration of at least 3 hours. This is valid for 56 cases, which are used for statistical analysis in the following sections.

The time derivative of the RLWP is estimated by calculating the slope of a linear fit on RLWP data within $\pm$ 30 minutes of a given time block. The retrieved slope value is declared valid only if at least 75% of the RLWP samples used in its calculation
are valid.

### 5.2.1    Fog formation

Figure 10 (a.1) shows the statistical behavior of RLWP between 90 minutes before and 90 minutes after for formation. It can be seen that at fog formation there is a transition from negative to positive RLWP values. The relatively lower amount of samples before -35 minutes from fog formation happen because there are less fog cases were the cloud has formed that early, or that
have an identifiable CTH above 85 meters. Yet, we can see that RLWP cannot be significantly lower than -10 g m$^{-2}$ if fog will form within 30 minutes.

Additionally, in Fig. 10 (a.2) we can see that dRLWP/dt becomes positive about one hour before formation, and remains consistently positive for another hour after formation. This first hour after fog formation is when fog reservoir grows the most, reaching a change rate of 10 to 25 g m$^{-2}$ Hr$^{-1}$, and it may be critical in establishing fog persistence for the coming hours.
After this first hour, fog RLWP stabilizes around 10 to 20 g m$^{-2}$ and the increase per hour is reduced until entering the mature stage.





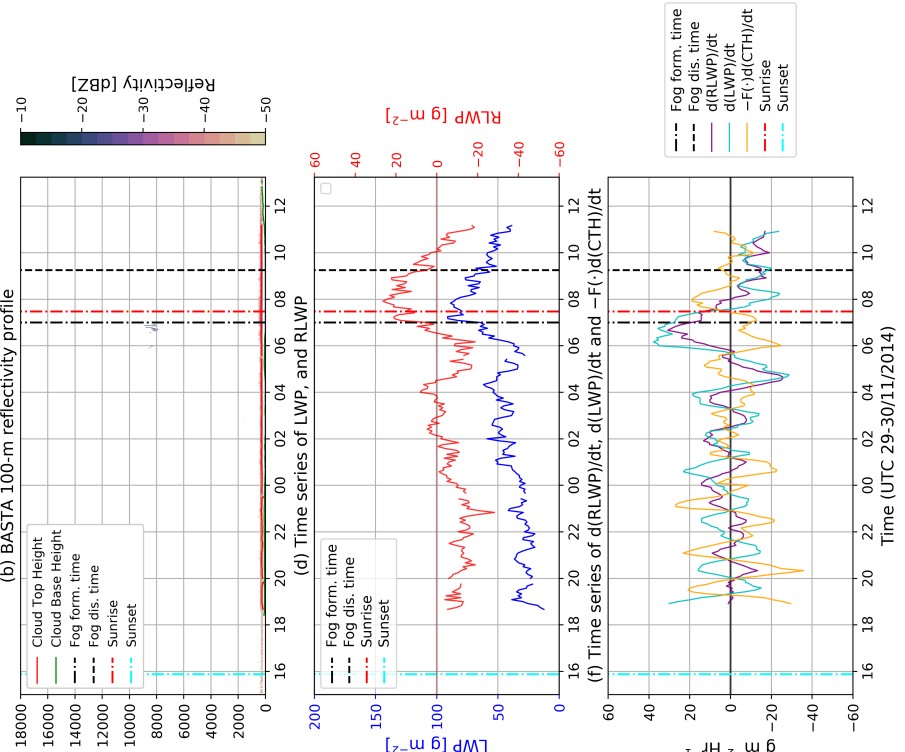

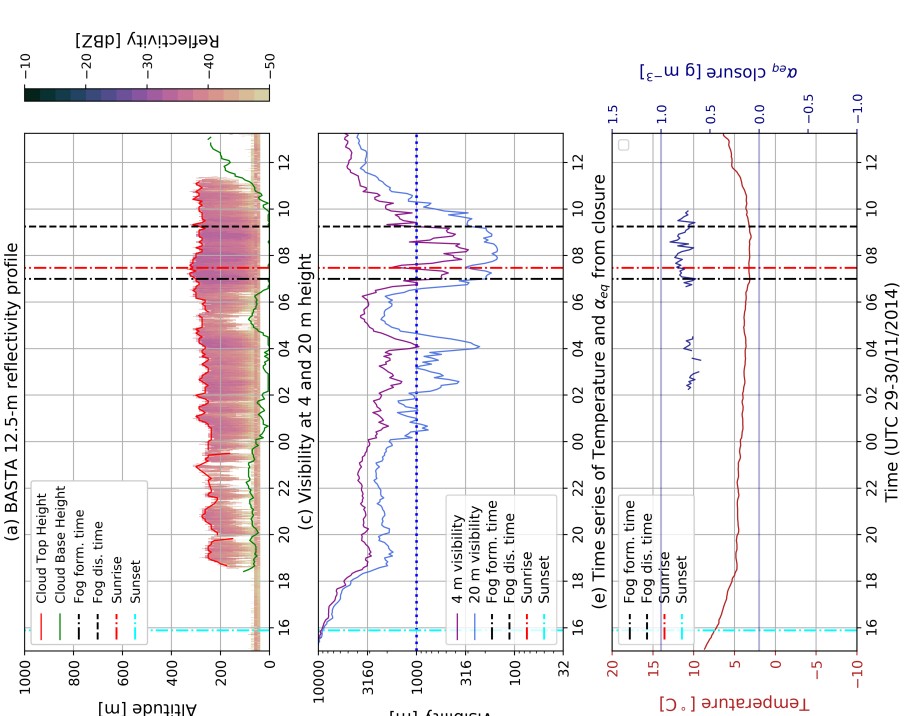

**Figure 9.** Case study 4. (a) Cloud Base Height (CBH), Cloud Top Height (CTH), and the cloud radar 12.5 m resolution reflectivity profile for the first 1000 m of height. (b) Cloud radar 100 m resolution reflectivity profile up to 10 km of height. (c) 4-m and 20-m horizontal visibilities. (d) Fog/Stratus layer measured LWP and computed RLWP. (e) Temperature and closure adiabaticity (calculated only when visibility is less than 2000 m). (f) Change rate of RLWP, with the individual contributions from LWP and CTH variations. In each panel, the time of fog formation and fog dissipation are clearly marked as well as the time of sunrise.





### 5.2.2 Fog mature stage

A histogram with RLWP values in the period defined as fog mature stage is shown in Fig. 10 (b.1). We can see that approx-
imately 90% of the time fog has a positive RLWP value, with a median value of 20.1 g m$^{-2}$ over all cases. Negative RLWP
values in fog mature stage are explained by short-term temporary lifting of fog from the surface, most likely caused by RLWP
oscilations.

Figure (b.2) shows the statistics of dRLWP/dt versus the sliding mean value of RLWP using the same data points involved the
slope calculation. This figure shows that RLWP and its time derivative are not correlated, and that most of the time dRLWP/dt
remains within $\pm$ 20 g m$^{-2}$ Hr$^{-1}$. The very low median value of dRLWP/dt = -0.2 g m$^{-2}$ Hr$^{-1}$ shows that fog does not have a
clear tendency of RLWP increase or decrease in the long term. Thus, during this stage of fog life cycle, RLWP remains positive
most of the time, with variations driven by oscillations in the value of dRLWP/dt.

### 5.2.3 Fog dissipation

In the latter stage of fog life cycle, shown in Fig. 11 (a.1), RLWP decreases consistently from positive values associated to the
middle of the life cycle until reaching negative values after fog dissipation. Figure 11 (a.2) shows that the monotonous decrease
in RLWP begins about 60 minutes before fog dissipation, and can commonly reach values of about -10 to -30 g $m^{-}2$ Hr$-1$.
These negative values in the time derivative continue after fog dissipation, and can be explained by further lifting or drying of
the remaining low stratus cloud (Wærsted et al., 2019).

To observe what is the main driver of fog dissipation, we calculate dRLWP/dt, dLWP/dt and $-F(CTH, \Gamma_{ad}, \alpha_{eq})\cdot$dCTH/dt,
defined in Sec. 4.3, using the last 60 minutes of data before dissipation. Theoretically, dissipation can only happen when the
RLWP decreases, which only happens when the sum of the LWP and CTH time derivative terms is negative (Eq. (8). This
matches the results of Fig. 11, which has most points in the quadrants leading to the aforementioned condition. The few points
that show a RLWP increase before dissipation, to the right of the dashed line, are associated with uncertain retrievals due to
low absolute RLWP values, or fast RLWP depletion in the few minutes just before dissipation (time trends are calculated using
a one hour linear fit).

Overall, data shows that fog dissipation happens in the same scenarios predicted by the Conceptual Model in Sec. 4.3.
Fog dissipates even when the LWP increases when the layer thickening effect is larger (large CTH increase), and leads to a
net reduction of RLWP. Alternatively, fog can also dissipate when LWP decreses, even if CTH subsides. Finally, some cases
dissipate with the contribution of both effects, LWP decrease and layer thickening. Data also shows that some regimes are
forbidden in the last 60 minutes of fog life cycle, specifically any case in which the addition of dLWP/dt and $-F(CTH, \Gamma_{ad},$
$\alpha_{eq})\cdot$dCTH/dt is positive. This condition can be helpful for identifying conditions where fog is likely to persist or dissipate.

**Figure 10.** The boxplots of panels (a.1) and (a.2) represent RLWP and dRLWP/dt statistics for each time block 90 minutes before and after fog formation. Boxplot shows the 25th, 50th and 75th percentiles, and the maximum and minimum values. Panels (b.1) and (b.2) show RLWP and dRLWP/dt statistics during fog middle life, between 90 minutes after fog formation and 90 minutes before fog dissipation. Only fog events longer than 3 hours are considered.





**Figure 11.** The boxplots of panels (a.1) and (a.2) show RLWP and dRLWP/dt statistics for each time block, 90 minutes before and after fog dissipation. Panel (b) uses the last 60 minutes of data before dissipation to calculate the impact of LWP and CTH variations in RLWP depletion. The dashed line indicates the theoretical limit where fog dissipation is possible (only to the left of this line). In quadrants II and III cloud base lifting contributes to RLWP decrease, while in Quadrants III and IV the LWP decrease contributes to RLWP depletion.





## 6   Conclusions

This work presents a Conceptual Model for adiabatic fog that relates fog liquid water path with its thickness, surface liquid water content and adiabaticity. The model predicts that LWP can be split into two contributions: the first is proportional to the adiabaticity and the square of CTH, and the second is the product of surface LWC and CTH. The later dependency is due to an excessive accumulation of water with respect to an equally thick cloud, which happens only in fog because the surface presence limits vertical development.

This excess accumulation of water motivates the definition of two parameters, which later will prove to be key in understanding fog evolution: the Critical LWP and the Reservoir LWP. The Critical LWP is the minimum amount of column water that would fill the fog layer and cause a visibility reduction down to 1000 m at the surface. The Critical LWP can be calculated using the conceptual model, by imposing a surface LWC equivalent to a 1000 m visibility. Meanwhile, the Reservoir LWP is the difference between fog LWP and the Critical value, and represents the excess of water that enables fog persistence. Case studies and statistical results show that the Reservoir LWP is positive when fog is present, and reaches 0 g m$^{-2}$ at about the same time as fog dissipation.

The model is used to statistically study fog adiabaticity. Important conclusions are that thinner fog has relatively lower adiabaticity values, less than the 0.6 value commonly found in literature for boundary layer clouds, even reaching negative values. This happens when the fog layer is not yet opaque during the fog formation stage, when LWC distribution is not even and may be larger closer to the surface. Meanwhile, when fog is developed, its adiabaticity value gets closer to previously observed values for boundary layer fog, converging at approximately 0.66 for fog thicker than 250 meters. An adiabaticity parametrization as a function of fog thickness is derived to perform conceptual model calculations.

Using this parametrization, the conceptual model enables an estimation of fog LWP with an RMSE of 10.5 g m$^{-2}$, which is close to the uncertainty in LWP measurement of 10 g m$^{-2}$. Additionally, data shows that the modeled LWP dependency on surface LWC, temperature and CTH is well captured by the model formulation.

The temporal derivative of the Reservoir LWP is studied, obtaining an analytic formulation that enables the separation of LWP and CTH contributions to reservoir depletion, and thus in causing fog dissipation. This formulation predicts that fog dissipation will depend on the relationship between LWP and CTH changes, and that it is possible for fog to dissipate even if LWP is increasing or CTH is decreasing if the other term impact on reducing the reservoir is larger. This prediction is verified by statistically studying the LWP and CTH dissipation trends, where in some cases it was found that dissipation happened under antagonist LWP and CTH effects.

In addition, the database is used to statistically observe the Reservoir LWP time derivative, finding a significant increase of Reservoir LWP about 60 minutes before and after fog formation. This is followed by oscillating Reservoir LWP variations of lower magnitude, sometimes created by larger but compensating LWP and CTH variations, sustaining a stable positive Reservoir LWP value during fog middle life. Fog life cycle ends when the Reservoir starts to decrease consistently, a trend that starts roughly 60 minutes before fog dissipation.





The aforementioned conclusions and the paper results indicate that the Reservoir LWP and its time derivative can be used
as indicators of fog life cycle stage and as a diagnostic tool to predict how close fog is from dissipation at the local scale.
Reservoir LWP forecasting is also conceivable by eventually including fog processes linked to LWP and CTH variations in the
calculations.

Regarding future work, it could be interesting to implement this framework on LES simulations, to improve our understanding of adiabaticity and of the surface effect in the excess accumulation of water that happens in fog with respect to equally
thick clouds.

Other area of future work is to improve the exploitation of the measurements. For example, the conceptual model formulation
could be tested with data obtained in other sites with frequent fog events, to test how general is the adiabaticity parametrization.
It would also be of interest to study how cloud radar reflectivity profiles could improve the quality of fog RLWP estimation.
For example by enabling the retrieval of adiabaticity profiles instead of relying on a single average value. Other observations
of interest include the quantification of local processes and larger area effects, such as advection, to enable the forecasting
of RLWP. RLWP forecasting would greatly improve our real time assessment of fog dissipation tendency. However, these
improvements would require a significant improvement on cloud radar calibration standards. As is explained in the introduction,
cloud microphysic retrievals are very sensitive to the radar calibration. Additionally, to enable the comparison of data between
different sites, standarized calibration methodologies must be put in place. Thus, significant work is done to rise solution
proposals for these issues. These proposals are presented in the following two chapters of the thesis.

*Data availability.* All data used in this study is hosted by the SIRTA observatory. Data access can be requested for free following the
conditions indicated in the SIRTA data policy (https://sirta.ipsl.fr/data_policy.html).

SIRTA observatory website: https://sirta.ipsl.fr/

Data request form: https://sirta.ipsl.fr/data_form.html

# Appendix A: Calculation of $\Gamma_{ad}(T, P)$

The inverse of the saturation mixing ratio change with height $\Gamma_{ad}(T, P)$ is calculated using the formulation published by
Albrecht et al. (1990) and Braun et al. (2018), shown in Eq. (A1).

$$\Gamma_{ad}(T, P) = \left[ \frac{(\epsilon + w_s)w_s l_v}{R_d T^2} \Gamma_w - \frac{g w_s P}{(P - e_s)R_d T} \right] \rho_d \tag{A1}$$

A description and the equations necessary to calculate each term used in the calculation of $\Gamma_{ad}(T, P)$ are given in Tab. A1.


**Table A1.** List of all the terms needed for the calculation of $\Gamma_{ad}(T,P)$.

| Term | Definition | Calculation | Units |
|---|---|---|---|
| $T$ | Surface temperature | | K |
| $P$ | Surface pressure | | Pa |
| $l_v$ | Latent heat of vaporization | $2.5 \cdot 10^6$ | J Kg$^{-1}$ K$^{-1}$ |
| $c_p$ | Specific heat of dry air at constant pressure | 1005 | J Kg$^{-1}$ K$^{-1}$ |
| $g$ | Acceleration of gravity | 9.81 | m s$^{-2}$ |
| $R_d$ | Dry air ideal gas constant | 287.0 | J Kg$^{-1}$ K$^{-1}$ |
| $R_v$ | Water vapor ideal gas constant | 461.5 | J Kg$^{-1}$ K$^{-1}$ |
| $\epsilon$ | Ratio of $R_d$ to $R_v$ | $\frac{R_d}{R_v}$ | |
| $e_s$ | Vapor saturation pressure | $611.2 \cdot \exp\left(\frac{17.67(T-273.15)}{T-29.65}\right)$ | Pa |
| $w_s$ | Saturation mixing ratio | $\epsilon \frac{e_s}{P-e_s}$ | |
| $\rho_d$ | Dry air density | $\frac{P-e_s}{R_d T}$ | Kg m$^{-3}$ |
| $\Gamma_w$ | Moist adiabatic lapse rate | $\frac{g}{c_p}\left(1+\frac{l_v w_s}{R_d T}\right)/\left(1+\frac{\epsilon l_v^2 w_s}{R_d c_p T^2}\right)$ | K m$^{-1}$ |
| $\Gamma_{ad}(T,P)$ | | Eq. (A1) | Kg m$^{-4}$ |

## Appendix B: Visibility-LWC parametrization

Surface LWC estimation from visibility measurements is done by inverting Gultepe et al. (2006) Eq. (6). This results in Eq. (B1), where LWC is Liquid Water Content in Kg m$^{-3}$ and VIS is the visibility in meters.

$$LWC = 0.0187 \cdot 10^{-3} \cdot \left(\frac{VIS}{1000}\right)^{-1.041} \tag{B1}$$

*Author contributions.* FT and MH developed the conceptual model and its formulation, based on initial work by EW and MH. FT and EW developed the code used for data analysis. FT and MH defined the paper structure and content. MH and JCD manage the SIRTA observatory, which provided the used dataset. All authors reviewed the paper.

*Competing interests.* The authors declare that they have no conflict of interest.

*Acknowledgements.* We acknowledge all the SIRTA observatory technical team for their extraordinary work on retrieving long term and high quality datasets. SIRTA measurements were performed in the framework of the ACTRIS, supported by the European Commission under the Horizon 2020 – Research and Innovation Framework Programme, H2020-INFRADEV-2019-2. We also acknowledge Marc-Antoine Drouin



and Cristophe Boitel of the SIRTA observatory for their help on data access. We acknowledge the french Association Nationale Recherche Technologie (ANRT) and the company Meteomodem for their contribution in the funding of this work.





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
