# Peer review of "A New Conceptual Model for Adiabatic Fog"

_Atmospheric Chemistry and Physics, 2020_

## Author Comment (AC1)

**Response to reviewer 1**

Dear reviewer, first we would like to thank you for the review and for your comments. They were very helpful in improving the readability and quality of the manuscript.

Below we present our detailed answer. Your comments are in green text, followed by our answers in black text. The referred lines in the answers are indicated with respect to the change tracking document, at the end of this file.

The authors present a conceptual model of fog water content profiles. They develop and evaluate it based on measurements, and use it to characterize fog life cycle stages.

**Remarks**

- line 1: Before you describe the details of the definition: Why is this new definition necessary?

Thanks for this observation. There is a need for better descriptions of fog dissipation. We propose a new way of looking at well-mixed or adiabatic continental fog as a layer that has a given top height and that is filled with liquid water. Dissipation occurs when the amount of liquid water becomes insufficient to fill the layer all the way to the ground. The first paragraph has been rewritten to provide a better introduction, introducing the problematic and the proposed solution globally. This new text is within lines 1-6.

- 2: the extent is first defined from the surface up, but in the following sentences the direction is inverted. Please find a consistent direction. Shouldn't it be "extends towards the surface from a known upper boundary altitude"?

To improve consistency, the direction of fog development is now always from the upper limit downwards. This is corrected in lines 7-14.

- 2: saturated by what?

By "saturated" we meant "with the presence of liquid water droplets due to water vapor saturation". The phrase was modified to better explain this point (lines 7-8).

- 17: would RLWP not be 0 by definition at dissipation?

From a theoretical point of view, this is correct. However, RLWP calculations are done using the theoretical model combined with empirical retrievals of CTH, LWP, Temperature and Pressure, and using an empirical parametrization to calculate adiabaticity.

Therefore, it was of interest to observe if real calculations were consistent with the theoretical principle. We now indicate explicitly that the statistics of the calculated RLWP values validate the predicted behavior from theory, and that RLWP is calculated using measurements and an empirical parametrization, in lines 23-25.

Abstract: The term 'adiabatic' fog is used here, and the assumption seems to be that fog has an adiabatic profile. This should be explicitly mentioned for clarity

Now this is mentioned explicitly in lines 7-8.

- 68: What precisely is meant by "rule" in this sentence? What are macroscopic properties in this context?

We improved the sentence. We now present our hypothesis more clearly, and explicitly state the macroscopic variables considered. This change can be seen in lines 78-81.

- 78: Why is opacity of a fog layer always linked to a well-mixed situation? Would you rule out the existence of an opaque sub-adiabatic fog layer? If so, why?

We do not rule out an opaque sub-adiabatic fog layer. The time required for a fog layer to become well-mixed once it is radiatively opaque will depend on the original stability and the abundance of cooling at fog top and heat fluxes from the surface. We have reformulated the text so it now says the fog layer will tend towards being adiabatic, in lines 90-95.

- 87: You say "the surface limits vertical fog development". But radiation fog development starts at the surface, and continues upwards. The implication of your statement would be development in the opposite direction. Is this intended? I would hold that "excess LWC" at the ground surface is the result of continuing cooling, not of a truncation of the cloud base.

Radiation fog does form close to the surface, and develops upwards as more water is condensed. This regime lasts as long as there is a stable temperature profile. However, when fog becomes opaque to infrared radiation, the main source of radiative cooling moves from the surface to the top of the fog layer. Surface radiative cooling is strongly reduced. The temperature profile transitions to neutral and the fog layer develops up to the temperature inversion. Eventually, LWC starts to mix vertically in the cloud and fog becomes adiabatic (e.g. Waersted 2017, Smith 2018). When fog is adiabatic, water vapor condensation happens predominantly at fog top (due to the radiative cooling), however, in spite of vertical mixing that occur due to the descent of colder air, the cloud cannot grow downward . because of the presence of the surface.

The lines in text were written thinking of this scenario, but after a revision we found that they may be confusing since not enough context was given. Therefore we rephrased lines 99-107 to improve the explanation of the phrase you cite.

- 93: Is this increase of LWC really always due to upwards motion of moisture, as you say, or could it also be upward movement of the cooling surface?

The left term of equation (2) is exclusively related with the saturation of ascending moist air. The right term (LWC0) is also included to account for direct LWC distribution to the column. This can be seen more clearly in eq. (3c), where we see that LWP is equal to a term related with moist air condensation plus a second term, which is the product of surface LWC (LWC0) and fog CTH. This second term is included to consider the redistribution of additional LWC generated by, for example, radiative cooling at the surface or at the fog top. The current formulation implies that this additional LWC is distributed approximately evenly in the fog column, which may not be physically exact, but it is a simplification we made for this conceptual model.

- 240: In your observations, does $\alpha_{eq}$ scale with visibility at ground level?

Below we attach a figure with the distribution of $\alpha_{eq}$ calculated from closure (based on observations) versus surface visibility.

This figure shows two different regimes. When fog is adiabatic, alpha_eq has values that are comparable with well mixed clouds for any surface visibility (between ~0.5-0.8). In this case alpha_eq does not scale with visibility.

When fog is not well mixed (alpha_eq << 0.5), it is also not possible to find a direct relationship between alpha_eq and visibility, because alpha_eq is highly variable. The reason for this variability is that in this regime, fog is either shallow (not well mixed), or starting to develop into an adiabatic layer, hence physical conditions are highly variable and cannot be directly estimated from surface visibility conditions alone.

[Figure]

- 473: What do you mean by implementation "on LES"? Implement your model in the LES?

We meant to use the output of LES simulations to test the conceptual model, and see if the behavior of the conceptual model variables on modeled fog was consistent with what is expected from theory. This is now better explained in lines 571-573.

- 477: While your findings/model are well backed up by the extensive data base of observations from the Paris region, to what extent do you expect your model to be applicable on other conditions? Can you comment on that, please?

Thanks for this comment, it is a very important topic. We now include in the conclusions a paragraph with our thoughts on the generality of the model. This paragraph can be read in lines 574-579.

**Details and technicalities**

- 1: For this purpose, fog is defined as...

- 21: water vapour saturation

- 21: 'cooling of air' or 'reduction of air temperature'

- various places: associated WITH

Thank you for your detailed review. All these technicalities are corrected.

- 484: These sentences probably don't belong here :) If the thesis is available online or via libraries, a reference would be useful to the interested reader, however.

Thank you for noticing this typo, indeed the sentences were misplaced. Unfortunately the thesis is not yet published, so at present it is not possible to add a citation. It should be available in a few months.

[revised manuscript text omitted]

---

## Author Comment (AC2)

Response to reviewer 2

Dear reviewer, first thank you for your encouraging comments. We sincerely appreciate your interest in this work. Additionally, we thank you for your comments, which were very helpful to improve the quality of the manuscript.

Below we present our detailed answer. Your comments are in blue text, followed by our answers in black text. The referred lines in the answers are indicated with respect to the change tracking document, at the end of this file.

**Summary :**

This work presents a conceptual model for adiabatic continental fog that relates fog liquid water path with its thickness, surface liquid water content and adiabaticity. This is an original powerful method to better understand fog life cycle and it could be the basis of a diagnostic tool for fog dissipation nowcasting.

Overall, I really like this work. It is the outgrowth of a good deal of soul-searching that began with Waersted's papers based on LWP budget. The manuscript is generally well written, and the figures in the manuscript are relevant. That being said, I would like the authors to be more upfront about the types of fog the model applies, the range of validity and the limits of the approach.

My concerns are presented below.

**General comments:**

1. The conceptual model is applied to fog formation and fog life cycle statistics are produced: does it concern only fog by stratus lowering? In this case, the sample size should be given and it should be clearly said that it only applies to fog by stratus lowering to remove the ambiguity (to be corrected at different locations and in the conclusion). If other types than the fog formed by stratus lowering are included, the conceptual model cannot be applied as the adiabaticity criteria is not verified for thin fogs. This would call into question the validity of the conceptual model.

Data from radiation and stratus lowering fog is considered in all the calculations. The only criteria used to detect fog events is visibility. This decision enabled adiabaticity retrievals for all fog types and situations, and enabled the finding of negative adiabaticity values when the LWP is less than ~30 g m-2. Negative adiabaticity indicates that fog LWC is, on average, decreasing with height, and therefore that fog is not adiabatic. The figure below, done using both fog types, complements this answer. The first plot shows the distributions of closure adiabaticity samples for all LWP ranges, the second for LWP values below 30 and the third for LWP values above 30 g m-2.

From this figure, we confirm that fog is adiabatic when the LWP is greater than 30 g m-2 (the limit where fog usually becomes opaque to infrared radiation, according to Waersted et al. (2017)). This enables the identification of the limits where fog lifting could not occur (fog can

lift only when it is adiabatic). This does not mean that we cannot do conceptual model calculations for other situations, but rather that the premise (lifting when the RLWP is 0 g m-2) will not necessarily be true under specific conditions.

It's also worth noting that when the LWP is above 30-40 g m-2, adiabaticity is about 0.6 for most samples, without distinction between radiative or stratus lowering fog. Therefore, rather than the fog type, what is key is to have an adiabatic layer to predict fog lifting using the conceptual model. Your comment motivated the introduction of a new figure 3 (b), showing the eq. adiabaticity vs LWP, and the separation between the sections analyzing adiabaticity results and the adiabaticity parametrization. As stated before, it was essential to apply the conceptual model to radiative and stratus lowering fog cases to identify these regimes. This discussion on the limits of applicability of the model is now developed in a much clearer way in Section 4.1 (lines 284-308), and in the conclusions (lines 522-531).

**Regarding the amount of samples**, several comments and a supplementary material have been added to provide more clarity.

In the specific case of fog formation, in general it was not possible to observe radiation fog before and at formation time because of its low CTH (below the 85 m minimum from the radar). Therefore, statistics should consider data related with stratus lowering cases mostly. This is now stated in lines 469-472.

Additional statements on the amount of cases and samples considered can be read on lines 469-472 and 483-485, and in the caption of figures 10 and 11.

[Figure]

2. Concerning fog dissipation, it can be caused by three different factors: the increase of heat surface fluxes, the presence of higher clouds and the advection of dry or warm air. Most of the time, only the first factor has a slow effect on the LWP evolution, and can be detectable a few hours before, while the other factors have a drastic impact. I have understood that the cases with presence of higher clouds are excluded from the study. What's about the dissipation due to large scale conditions: are they also excluded and what is the criteria to exclude them? What are the statistics about the factors of dissipation at the SIRTA site? In Waersted et al. (2019) it was written that more than half the fog events dissipate after sunrise transition to a stratus which lasts at least 2 h. It is important to remind these statistics for the SIRTA site and to better define when the conceptual model can be applied to dissipation cases and to present the limits of the approach.

In principle, all cases are considered, since they are identified based on surface visibility measurements only. What is done, is that each 5 min average block of LWP samples is declared invalid if liquid clouds are suspected above the fog layer, since that would introduce overestimated fog LWP measurements. This is now better explained in lines 233-248.

This will, of course, reduce the amount of data contributed by fog cases that dissipated by the radiative sheltering due to the presence of higher clouds, however these cases are not excluded manually or following any specific criteria. The same applies for the other dissipation scenarios.

Yet, we think that this should not present an issue, because the conceptual model provides a diagnostic variable (the RLWP) which is not a function of the processes acting on fog at any given time, but rather on fog geometry and liquid water content.

However, we also agree that our study only considers continental fog at SIRTA. The results should be rather general for these fog types, since the results match previous observations of adiabaticity (lines 296-299). The applicability of the conceptual model to marine fog for example should be studied. This is now discussed in the article conclusions (lines 574-579).

As explained in the previous comment, the limits of the approach are discussed in lines 284-308 and 522-531.

3. The approach does not take into account the droplet concentration (Nc). But we know that for a same liquid water content (LWC), a fog will be optically thicker if Nc is high and vice-versa. It lasts also longer due to the droplet sedimentation which is reduced. In the same way, Dupont et al. (2012) have shown that evaporation of the droplets below the stratus is one of the main factors contributing to stratus lowering and fog formation, so a small concentration of droplets favors the growth of the droplets, their sedimentation and evaporation. How would you introduce these considerations in the approach? Do you consider that the impact of the microphysics is included in the LWP evolution, or that it is of $2^{nd}$ order compared to the LWP and CTH evolutions? Again it is important to introduce this point of discussion.

LWP, CTH and adiabaticity drive the conceptual model variables. The equivalent adiabaticity should be related to fog microphysics, yet at present this relationship is not studied. Droplet number concentration may also have an impact on the calculation of surface LWC versus visibility, and therefore not considering it probably increases uncertainty when calculating fog LWP from the conceptual model. Its impact could be studied using the output from LES to study the behavior of the RLWP and of fog adiabaticity for different microphysical conditions. This is now better explained in the conclusions (lines 571-573).

Additionally, the conceptual model provides instant diagnostic variables about fog status at a given time, yet the main factors that you mention are processes that drive fog evolution. These mentioned processes and conditions could be used to predict the evolution of fog LWP, and also to estimate the evolution of the RLWP (paired with other processes to estimate fog CTH development and Temperature changes). This is now discussed in the conclusions (lines 560-568).

4. The readability of the paper could be improved if all the formula/equations were grouped together in one initial part when the fog conceptual model is presented.

We agree with the reviewer that a section presenting all the equation would help the readability of the paper. However, the constraint is  that some equations arise from theory (section 2), and some are based on empirical observations (section 4). Therefore, it is complicated to introduce all the equations in the same place. In the end we found out that our original distribution works best in the article, because it is necessary to (1) to introduce the theory to explain which data is important, and (2) to describe the dataset and how it is processed, before performing calculations and obtaining the empirical equations.

5. The different cases are presented as a catalog, with the first three illustrating a dissipation by stratus, except the last one which considers a fog by stratus lowering. Could you introduce the benefit to present the second and the third cases, as only the first and the fourth should be kept. Or you cloud present another type of fog, for instance an advective fog, to better discriminate the limits of the approach.

We revised the cases, and in the end decided to leave cases one, two and four. Case three was moved to the supplementary material.

In case "one" fog dissipates in the morning due to LWP depletion. Meanwhile, in case "two", fog dissipates by CTH lifting. These two cases remain in the paper because they illustrate dissipation by different mechanisms.

The previous case "three" was an example of a long-lasting fog case, probably explained by a very high RLWP value at sunrise, and strong thermal inversion at fog top preventing CTH to increase significantly after sunrise. This may be interesting on its own, however we agree that it does not introduce much more significant elements for analysis.

Case "four" shows an application of the conceptual model to stratus-lowering fog, therefore it remains interesting and different to the previous cases, therefore as you propose we did not remove it.

**Detailed comments**:

1. l 32 : "An adiabatic fog behaves similarly": *almost* must be added because eddies are smaller in fog than in a stratocumulus.

Similarly in this context means, "alike yet not exactly the same", so in the end we decided to keep it as it is.

2. l 33: "stratocumulus clouds" must be replaced by "adiabatic fogs" as Nakanishi, 2000; Porson et al., 2011; Bergot, 2013, 2016; Wærsted et al., 2019 have not studied stratocumulus.

Thank you, we changed this in line 43.

3. l 172: As you have applied the Tardif and Rasmussen (2007) identification algorithm, how are partitioned the fog types, between radiative fog, fog by stratus lowering and advective fogs? Do you keep all the types in the rest of the study?

We have not classified or separated between fog types. We consider all fog events that are identified based on the Tardif and Rasmussen algorithm, using surface visibility measurements only. Since in this study we are aiming to find general properties, we prefered to use as much data as possible. The details on how data is selected and used are explained in our answer to the general comments 1 and 2.

4. l 206: It is not clear if cases with higher clouds detected by the radar are excluded.

The cases are considered. What is excluded is the LWP retrieval, and therefore the conceptual model calculations for the given 5 min time blocks associated with higher clouds. This is explained in our answer to the general comment 2, and in lines 233-241 and 244-248 of the text.

5. Fig. 4: What are the heights of temperature and visibility measurements?

The heights of the instruments are now presented more clearly in Table 1.

6. First case study: how do you use the fact that a is much more different than 0,66 after 7 UTC?

After sunrise, between 7-9 UTC, LWP decreases significantly (-25 gm-2) while the near-surface visibility (or LWC0) remains rather constant and CTH decreases moderately. Equation 6 (and Fig 7d) is the closure equivalent adiabaticity. It reveals if changes in LWP, CTH and LWC0 are not consistent with each other through changes in alpha-closure. Hence, here as alpha-closure deviates from the adiabatic value of 0.66 between 7-9 UTC, one could conclude that CLWP and RLWP calculations are less realistic then if alpha-closure remained near 0.66.

7. Fig.6,7,8,9: (a): this subfigure could be zoomed up to 500m height. As there is no cloud above, what is the subfigure (b) for? (c) : can you move the legend insert? (e) what is the height of temperature measurement? (f) the yellow does not go well. It would be nice to improve its readability by increasing its size as it is the most important subfigure.

We now significantly improved the figure considering your recommendations.

8. l 349: "When there is strong cooling at the fog layer top, LWP increases and vertical circulation is intensified.": increased would be better than intensified as vertical velocity intensity is low. Where does this remark come from: is it a general consideration or the result of a figure which is not shown?

It is a general consideration, based on the cited work (E. Waersted 2018 thesis). We also replaced intensified by increased as recommended.

9. §5.2.1: Does it concern only stratus lowering? How many cases do you consider?

As explained in the answer to the general comment 1, this is now stated in lines 469-472.

10. §5.2.2: How many cases do you consider in the statistics? Idem for §5.2.3

The statistics of sections 5.2.2 and 5.2.3 consider 56 fog cases. All fog events are superimposed and considered for all calculations. This is done to look for general behaviors. However, as explained in Section 3.3, there may be a different number of valid samples contributed by each fog event, and this implies that each bin may be calculated using a different amount of samples.

To provide the number of samples per bin, we include two new figures in S2 of the supplementary material showing the number of samples for each bin.

11. Fig.10b1: Is the ordinate axis the same for the cumulative and norm. distributions?

Yes, it is the same axis. This is now stated in the caption of this Figure.

12. Fig.11: It would be interesting to plot the same kind of representation than Fig.10b2

Figure 10b2 shows the statistics for all samples in the fog middle life. We cannot do the same figure for all the samples in the dissipation phase. What we did, however, is to plot this figure for the last 60 minutes of fog, shown in the figure below (mean and derivative of data from the last 60 min before dissipation).

As you can observe, in the last 60 minutes the RLWP is in average positive, below 30 g m-2, and has a negative dRLWP/dt. There is no apparent correlation between the values of both variables.

Hence, this figure does not add much information with respect to current Fig. 11 (a.1) and (a.2). Yet, an extra line has been added in  (lines 488-490 and 562-563) to indicate that RLWP is very rarely above ~30 g m-2 in the last ~30 minutes before dissipation.

[Figure]

13. l 412: What does "using the same data points involved the slope calculation" mean?

This is now better explained. Lines 455-457 indicate in the same place how the sliding mean and the slope calculation is done.

14. l 427-430 434-435: Are you not talking about the same thing?

Yes, they are observations on the same figure and subject. Thanks for spotting the redundancy. The paragraph is now corrected in lines 501-506.

15. l 470: The model is presented as "a diagnostic tool to predict how close fog is from dissipation at the local scale". What are you aiming for and for what kind of user: to predict the dissipation in the next hour for airports? This point could be slightly more developed.

Thanks for helping to improve our conclusions. This is now more developed in lines 557-559.

16. What do you mean by "implement this framework on LES simulations"? A LES does not need a conceptual model as it resolves most of the processes. But you can use the results from LES to validate and improve your conceptual model.

We agree the phrase was not correct. We meant what you explain in your comment. This is now corrected in the conclusions (lines 571-573).

17. l 485: "These proposals are presented in the following two chapters of the thesis." must be replaced by a reference to Toledo et al. (2020) in AMT.

This was a mistake. In fact it was citing the same thesis chapter where the paper was on. Unfortunately we can't cite the related thesis in this work because it is not yet published, so this statement was removed.

**Minor corrections**:

- l 215 : consist**s**

- l 369: an LWP

- l 387 and 399: w**h**ere

- l 432: decre**a**ses

- l 414 and 420 : $h^{-1}$ instead of $Hr^{-1}$

Thank you for the detailed corrections, these typos have been fixed in the text and figures.

[revised manuscript text omitted]

---

## Author Response (AR2)

Dear Editor,

The requested changes are now included in the manuscript. Since the comments were only a few technicalities in the text, instead of a point-by-point response we just attach the file with change tracking (there are no relevant changes).

Thank you for your time,

Felipe Toledo